# Disease consequences of higher adiposity uncoupled from its adverse metabolic effects using Mendelian randomisation

Susan Martin[1], Jessica Tyrrell[1], E Louise Thomas[2], Matthew J Bown[3,4], Andrew R Wood[1], Robin N Beaumont[1], Lam C Tsoi[5], Philip E Stuart[5], James T Elder[5,6], Philip Law[7], Richard Houlston[7], Christopher Kabrhel[8,9], Nikos Papadimitriou[10], Marc J Gunter[10], Caroline J Bull[11,12,13], Joshua A Bell[11,12], Emma E Vincent[11,12,13], Naveed Sattar[14], Malcolm G Dunlop[15,16], Ian PM Tomlinson[17], Sara Lindström[18,19], INVENT consortium , Jimmy D Bell[2], Timothy M Frayling[1]*, Hanieh Yaghootkar[1,2,20]*

[1]Institute of Biomedical and Clinical Science, University of Exeter Medical School, Research, Innovation, Learning and Development building, Royal Devon & Exeter Hospital, Exeter, United Kingdom; [2]Research Centre for Optimal Health, School of Life Sciences, University of Westminster, London, United Kingdom; [3]Department of Cardiovascular Sciences, University of Leicester, Leicester, United Kingdom; [4]NIHR Leicester Biomedical Research Centre, Leicester, United Kingdom; [5]Department of Dermatology, University of Michigan, Ann Arbor, United States; [6]Ann Arbor Veterans Affairs Hospital, Ann Arbor, United States; [7]The Institute of Cancer Research, London, United Kingdom; [8]Department of Emergency Medicine, Massachusetts General Hospital, Boston, United States; [9]Department of Emergency Medicine, Harvard Medical School, Boston, United States; [10]Nutrition and Metabolism Branch, International Agency for Research on Cancer, Lyon, France; [11]MRC Integrative Epidemiology Unit at the University of Bristol, Bristol, United Kingdom; [12]Population Health Sciences, Bristol Medical School, University of Bristol, Bristol, United Kingdom; [13]School of Cellular and Molecular Medicine, University of Bristol, Bristol, United Kingdom; [14]Institute of Cardiovascular and Medical Sciences, University of Glasgow, Glasgow, United Kingdom; [15]University of Edinburgh, Edinburgh, United Kingdom; [16]Western General Hospital, Edinburgh, United Kingdom; [17]Edinburgh Cancer Research Centre, IGMM, University of Edinburgh, Edinburgh, United Kingdom; [18]Department of Epidemiology, University of Washington, Seattle, United States; [19]Division of Public Health Sciences, Fred Hutchinson Cancer Research Center, Seattle, United States; [20]Centre for Inflammation Research and Translational Medicine (CIRTM), Department of Life Sciences, Brunel University London, Uxbridge, United Kingdom

*For correspondence:
t.m.frayling@exeter.ac.uk (TMF);
Hanieh.Yaghootkar@brunel.ac.uk (HY)

## Abstract

**Background:** Some individuals living with obesity may be relatively metabolically healthy, whilst others suffer from multiple conditions that may be linked to adverse metabolic effects or other factors. The extent to which the adverse metabolic component of obesity contributes to disease compared to the non-metabolic components is often uncertain. We aimed to use Mendelian randomisation (MR) and specific genetic variants to separately test the causal roles of higher adiposity with and without its adverse metabolic effects on diseases.

**Methods:** We selected 37 chronic diseases associated with obesity and genetic variants associated with different aspects of excess weight. These genetic variants included those associated with metabolically 'favourable adiposity' (FA) and 'unfavourable adiposity' (UFA) that are both associated with higher adiposity but with opposite effects on metabolic risk. We used these variants and two sample MR to test the effects on the chronic diseases.

**Results:** MR identified two sets of diseases. First, 11 conditions where the metabolic effect of higher adiposity is the likely primary cause of the disease. Here, MR with the FA and UFA genetics showed opposing effects on risk of disease: coronary artery disease, peripheral artery disease, hypertension, stroke, type 2 diabetes, polycystic ovary syndrome, heart failure, atrial fibrillation, chronic kidney disease, renal cancer, and gout. Second, 9 conditions where the non-metabolic effects of excess weight (e.g. mechanical effect) are likely a cause. Here, MR with the FA genetics, despite leading to lower metabolic risk, and MR with the UFA genetics, both indicated higher disease risk: osteoarthritis, rheumatoid arthritis, osteoporosis, gastro-oesophageal reflux disease, gallstones, adult-onset asthma, psoriasis, deep vein thrombosis, and venous thromboembolism.

**Conclusions:** Our results assist in understanding the consequences of higher adiposity uncoupled from its adverse metabolic effects, including the risks to individuals with high body mass index who may be relatively metabolically healthy.

**Funding:** Diabetes UK, UK Medical Research Council, World Cancer Research Fund, National Cancer Institute.

## Editor's evaluation

The authors have conducted a robust and very comprehensive study using Mendelian randomisation to disentangle metabolic and non-metabolic effects of overweight on a long list of disease outcomes. They have tested if effects of overweight work through either or both effects for a particular condition. This is an important topic and can help us better understand how overweight influences risk of several important outcomes.

## Introduction

Obesity is associated with a higher risk of many diseases, notably metabolic conditions such as type 2 diabetes, but many individuals are often relatively metabolically healthy compared to others of similar body mass index (BMI). Whilst these metabolically healthier individuals may be at lower risk of some obesity-related conditions, they may be at risk of conditions that are linked to other aspects of obesity, such as the load-bearing effects. The burden of obesity on individuals and health-care systems is very large, and in the absence of a widely applicable, sustainable treatment or effective public health measures, it is important to understand the disease consequences of obesity, and how they may be best alleviated, in more detail.

To better understand the disease consequences of obesity, many previous studies have used the approach of Mendelian randomisation (MR) (*Smith and Ebrahim, 2004*). These studies used common genetic variants robustly associated with BMI as proxies for obesity to assess the causal effects of higher BMI on many diseases. MR studies have provided strong evidence that higher BMI leads to osteoarthritis (*Tachmazidou et al., 2019*), colorectal cancer (*Thrift et al., 2015*; *Suzuki et al., 2021*; *Bull et al., 2020*), and psoriasis (*Budu-Aggrey et al., 2019*), as well as metabolic conditions such as type 2 diabetes, cardiovascular disease (*Hägg et al., 2015*), and heart failure (*Cheng et al., 2019*; *Corbin et al., 2016*; *Fall et al., 2013*). Other MR studies indicate that higher BMI may lead to lower risk of some diseases, including postmenopausal breast cancer (*Guo et al., 2016*) and Parkinson's disease (*Noyce et al., 2017*).

Obesity is heterogeneous – for example, for a given BMI, people vary widely in their amount of fat versus fat free mass, predominantly muscle, and their distribution of fat, predominantly subcutaneous versus ectopic and upper versus lower body fat. Even when there is strong evidence of causality, obesity may lead to disease through a variety of mechanisms. Despite many MR studies testing the role of higher BMI in disease, few have attempted to separate and test the different mechanisms that could lead from obesity to disease. Some MR studies have investigated the effects of fat distribution using genetic variants associated with waist-hip ratio (WHR) adjusted for BMI and shown that adverse

fat distribution (more upper body, less lower body) leads to higher risk of metabolic disease (*Emdin et al., 2017*), some cancers (*Cornish et al., 2020*), and gastro-oesophageal reflux disease (*Green et al., 2020*).

Previous studies have identified genetic variants associated with more specific measures of adiposity. For example, several studies have characterised variants associated with 'favourable adiposity' (FA) or reduced adipose storage capacity using a variety of approaches (*Ji et al., 2019*; *Lotta et al., 2017*; *Kilpeläinen et al., 2011*; *Huang et al., 2021*). We recently identified 36 FA alleles which are collectively associated with a favourable metabolic profile, higher subcutaneous fat but lower ectopic liver fat (*Ji et al., 2019*; *Martin et al., 2021*), resembling a polygenic phenotype opposite to lipodystrophy (*Semple et al., 2011*). We also identified 38 unfavourable adiposity (UFA) alleles which are associated with higher fat in subcutaneous and visceral adipose tissue, and higher ectopic liver and pancreatic fat (*Ji et al., 2019*; *Martin et al., 2021*), resembling monogenic obesity (*Supplementary file 1a*). We performed MR studies and showed that FA and UFA have opposite causal effects on six metabolic conditions (*Martin et al., 2021*). While both FA and UFA were associated with higher adiposity, FA was causally associated with lower risk of type 2 diabetes, heart disease, hypertension, stroke, polycystic ovary syndrome, and non-alcoholic fatty liver disease. In contrast, as expected, UFA was associated with higher risk of these conditions. These results confirmed the ability of the two sets of adiposity variants to partially separate out the metabolic from the non-metabolic effects of higher adiposity.

In this study, we aimed to investigate the effects of separate components to higher adiposity on risk of additional metabolic diseases and many non-metabolic diseases. We used genetic variants associated with BMI, body fat percentage, FA, and UFA to understand the components of higher adiposity that are the predominant causes of disease risk. Our findings may give guidance on some obesity-related risks which are not dependent on metabolic consequences, thereby guiding appropriate medical care.

## Methods
### Study design
An overview of our approach is shown in *Figure 1*. First, we identified diseases by performing a literature search of studies that had used MR to assess the consequences of BMI on outcome phenotypes. We used the search terms 'BMI and Mendelian randomisation' and 'BMI and Mendelian randomization'. We identified 37 diseases associated with BMI and for which MR studies had previously been performed (*Supplementary file 1b*). We included all diseases regardless of the MR result in the published study. Second, we reperformed MR studies using BMI as an exposure. Third, for those diseases where MR indicated higher BMI was causal, we tested the effects of body fat percentage to confirm that the causal effect was due to fat mass rather than fat-free mass. Fourth, for diseases where MR suggested the BMI effect was due to excess adiposity, we used genetic variants more specific to the metabolic and non-metabolic components of higher adiposity to help understand the extent to which these factors influence disease.

### Data sources
We used three data sources for disease outcomes: (i) published genome-wide association studies (GWAS; *Okada et al., 2014*; *Nikpay et al., 2015*; *Jones et al., 2017*; *Michailidou et al., 2017*; *Phelan et al., 2017*; *Scelo et al., 2017*; *Tsoi et al., 2017*; *Day et al., 2018*; *Mahajan et al., 2018*; *Malik et al., 2018*; *O'Mara et al., 2018*; *Roselli et al., 2018*; *Schumacher et al., 2018*; *Wray et al., 2018*; *An et al., 2019*; *Ferreira et al., 2019*; *Huyghe et al., 2019*; *Jansen et al., 2019*; *Kunkle et al., 2019*; *Law et al., 2019*; *Lindström et al., 2019*; *Morris et al., 2019*; *Nalls et al., 2019*; *Shah et al., 2019*; *Tachmazidou et al., 2019*; *Tin et al., 2019*; *Wuttke et al., 2019*; *Huyghe et al., 2021*) and (ii) FinnGen (*FinnGen, 2021*) as our main results, and (iii) UK Biobank (RRID:SCR_012815; *Collins, 2012*) as additional validation. FinnGen is a cohort of 176,899 individuals with linked medical records. UK Biobank is a population cohort of >500,000 individuals aged 37–73 years recruited between 2006 and 2010 from across the UK. For the 37 identified diseases, 25 had summary GWAS data available from both a published GWAS consortium and FinnGen, and 12 diseases had GWAS summary data available in FinnGen only. In addition, data from 31 of the 37 diseases were available in the UK Biobank. No GWAS data were available for Barrett's oesophagus, but we included gastro-oesophageal reflux.

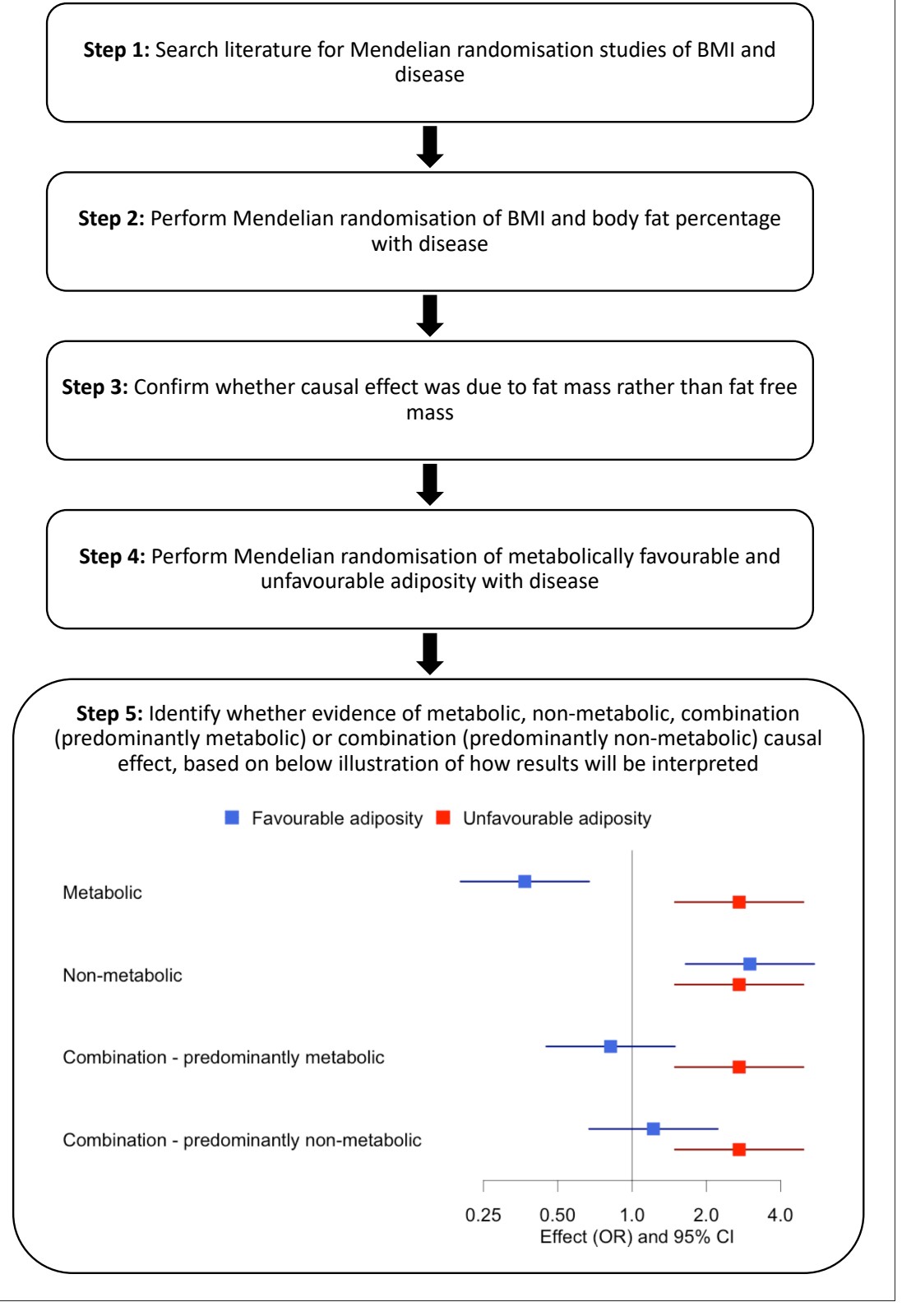

**Figure 1.** Study design.

The characteristics of the studies and measures, disease outcomes, and the definition of cases and controls are described in *Supplementary file 1ci–iii*.

## GWAS of UK Biobank participants

For the GWAS of 31 diseases available in UK Biobank, we used a linear mixed model implemented in BOLT-LMM to account for population structure and relatedness (*Loh et al., 2015*). We used age, sex, genotyping platform, study centre, and the first five principal components as covariates in the model.

## Genetic variants

We used four sets of genetic variants as proxies of four exposures (*Supplementary file 1d*).

### Body mass index

In the broadest category, we used a set of 73 variants independently associated with BMI at genome-wide significance ($p < 5 \times 10^{-8}$). These variants were identified in the GIANT consortium of up to 339,224 individuals of European ancestry (*Locke et al., 2015*).

### Body fat percentage

We used 696 variants from a GWAS in the UK Biobank (*Martin et al., 2021*). We used bio-impedance measures of body fat % taken by the Tanita BC-418MA body composition analyser in 442,278 individuals of European ancestry.

The BMI and body fat percentage variants were partially overlapping (n = 5 variants), but we used exposure-trait-specific weights for each variant.

### FA variants

There are 36 FA variants (*Martin et al., 2021*). These variants were identified in two steps. First, they were associated (at $p < 5 \times 10^{-8}$) with body fat percentage and a composite metabolic phenotype consisting of body fat percentage, HDL-cholesterol, triglycerides, SHBG, alanine transaminase, and aspartate transaminase. Second, in a k-means clustering approach (a hard clustering approach) (*Martin et al., 2021*), they formed a cluster of variants that were collectively associated with higher HDL-cholesterol, higher SHBG, and lower triglycerides and liver enzymes – resembling a phenotype opposite to lipodystrophy.

### UFA variants

There are 38 UFA variants (*Martin et al., 2021*). These variants were identified in two steps. First, they were associated (at $p < 5 \times 10^{-8}$) with body fat percentage and a composite metabolic phenotype as detailed above. Second, in a k-means clustering approach (*Martin et al., 2021*), they formed a cluster of variants that were collectively associated with lower HDL-cholesterol, lower SHBG, and higher triglycerides and liver enzymes - resembling monogenic obesity.

## Mendelian randomisation

We investigated the causal associations between the four exposures (BMI, body fat percentage, FA, and UFA) and 37 disease outcomes by performing two-sample MR analysis (*Pierce and Burgess, 2013*). We used the inverse-variance weighted (IVW) approach as our main analysis, and MR-Egger and weighted median as sensitivity analyses in order to detect and partially account for unidentified pleiotropy of our genetic instruments. For BMI, we used effect size estimates from the GWAS of BMI (*Locke et al., 2015*), and for body fat percentage, FA, and UFA, we used effect size estimates from the GWAS of body fat percentage (442,278 European ancestry individuals from the UK Biobank study) (*Ji et al., 2019*).

To estimate the effects of variants on our disease outcomes, we used two main sources of data: FinnGen GWAS summary results and published GWAS of the same diseases (*Supplementary file 1ci–ii*). We performed MR within each data source and then meta-analysed the results across the two datasets using a random-effects model with the R package *metafor* (RRID:SCR_003450; *Viechtbauer, 2010*), where the data was available in both. For one published GWAS (the GECCO consortium), we only had information for FA and UFA variants. To provide further MR evidence, we used a third source of disease data – disease status in the UK Biobank (*Supplementary file 1ciii*). We ran the same

models but did not meta-analyse with published GWAS and FinnGen because most of the body fat percentage, FA, and UFA variants were identified in the UK Biobank.

We obtained heterogeneity Q statistics for each IVW MR and MR-Egger, and $I^2$ statistics for each MR-Egger analysis using the *MendelianRandomization* R package (*Yavorska and Burgess, 2017*). All statistical analyses were conducted using R software (*R Development Core Team, 2020*). Given the number of tests performed, we used a Benjamini–Hochberg false discovery rate (FDR) procedure and an FDR of 0.1 to define meaningful results for each of the four exposures (*Benjamini and Hochberg, 1995*).

## Results

We identified 37 diseases as associated with obesity and for which MR studies had previously been performed. Of these 37, 5 metabolic conditions were part of our previous study that validated the use of FA and UFA genetic variants as a way of partially separating the metabolic from non-metabolic components of higher adiposity (*Martin et al., 2021*). Once we had tested BMI and body fat percentage, we further characterised the likely causal component of higher adiposity using FA and UFA variants as follows (*Figure 1*, step 5): (i) diseases with evidence that the metabolic effect of higher adiposity is causal. Here, MR using the UFA genetic variants indicated that higher adiposity with its adverse metabolic consequences was causal to disease, whilst MR using the FA genetic variants indicated that higher adiposity with favourable metabolic effects was protective (at FDR 0.1). (ii) Diseases with evidence that there is a non-metabolic causal effect (e.g. mechanical effect, psychological/adverse social effect). Here, MR using the FA genetic variants indicated that higher adiposity without its adverse metabolic consequences was likely contributing to the disease, as well as the MR using the UFA genetic variants. (iii) Diseases with evidence that there is a combination of causal effects but with a predominantly metabolic component. Here, MR using the UFA genetic variants indicated that higher adiposity with its adverse metabolic consequences was causal to disease, and MR using the FA genetic variants was directionally consistent with higher adiposity with favourable metabolic effects being protective but FDR > 0.1. (iv) Diseases with evidence that there is a combination of causal effects but with a predominantly non-metabolic component. Here, MR using the UFA genetic variants indicated that higher adiposity without its adverse metabolic consequences was likely contributing to the disease, and MR of the FA genetic variants was directionally consistent with this but FDR > 0.1.

We grouped these disease outcomes into seven major categories – cardiovascular and metabolic conditions, musculoskeletal, gastrointestinal, nervous, integumentary and respiratory systems, and cancer. MR analysis of five conditions (coronary artery disease, hypertension, stroke, type 2 diabetes, and polycystic ovary syndrome) was part of our previous study (*Martin et al., 2021*). We focused on the MR of body fat percentage if a causal effect of BMI was indicated, and the MR of FA and UFA if a causal effect of BMI and body fat percentage was indicated, but have presented all results in *Supplementary file 1e* for completeness. Where random-effects meta-analyses were performed, the heterogeneity statistics are given in *Supplementary file 1f*.

### (i) Diseases with evidence that the metabolic effect of higher adiposity is causal

When comparing the MR analyses for FA and UFA, our results provided evidence that the metabolic effect of higher adiposity is contributing causally to coronary artery disease, peripheral artery disease, hypertension, stroke, type 2 diabetes, and gout (*Figures 2–12*, *Supplementary file 1e*). For stroke, our results were consistent when using sub-types of the condition (*Figure 3—figure supplement 1*, *Supplementary file 1g*). Our results also indicated that the metabolic effect of higher adiposity is causal to chronic kidney disease, although the results from BMI and body fat percentage were less conclusive (*Figure 3*).

### (ii) Diseases with evidence that there is a non-metabolic causal effect

When comparing the MR analyses for FA and UFA, our results provided evidence that some non-metabolic effect of higher adiposity is contributing causally to venous thromboembolism, deep vein thrombosis, osteoarthritis, and rheumatoid arthritis (*Figures 2–12*, *Supplementary file 1e*). For

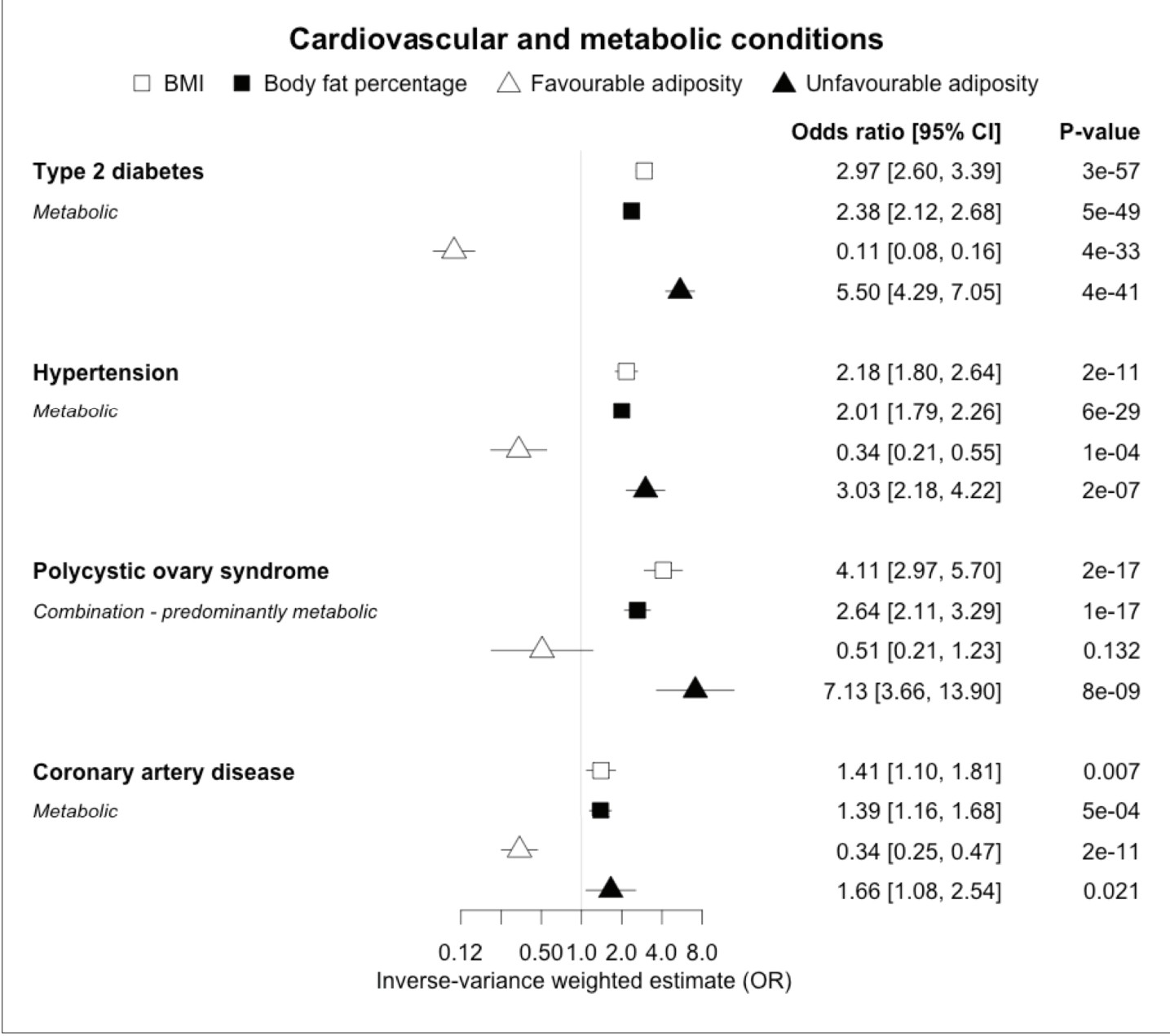

**Figure 2.** The inverse-variance weighted (IVW) two-sample MR analysis/meta-analysis of the effects of body mass index (BMI), body fat percentage (BFP), "favourable adiposity" (FA) and "unfavourable adiposity" (UFA) on type 2 diabetes, hypertension, polycystic ovary syndrome and coronary artery disease. The error bars represent the 95% confidence intervals of the IVW estimates in odds ratio per standard deviation change in genetically determined BMI, body fat percentage, FA and UFA. *Italics give our best interpretation of the data using the FDR 0.1 results.*

osteoarthritis, our results were consistent when using sub-types of the condition (*Figure 5—figure supplement 1*, *Supplementary file 1g*).

### (iii) Diseases with evidence that there is a combination of causal effects but with a predominantly metabolic component

When comparing the MR analyses for FA and UFA, our results provided evidence that the metabolic effect of higher adiposity is the predominate cause of the link between higher BMI and polycystic ovary syndrome, heart failure, and atrial fibrillation. Our results also provided evidence that the metabolic effect of higher adiposity is the predominate cause of the link between higher BMI and a reduced risk

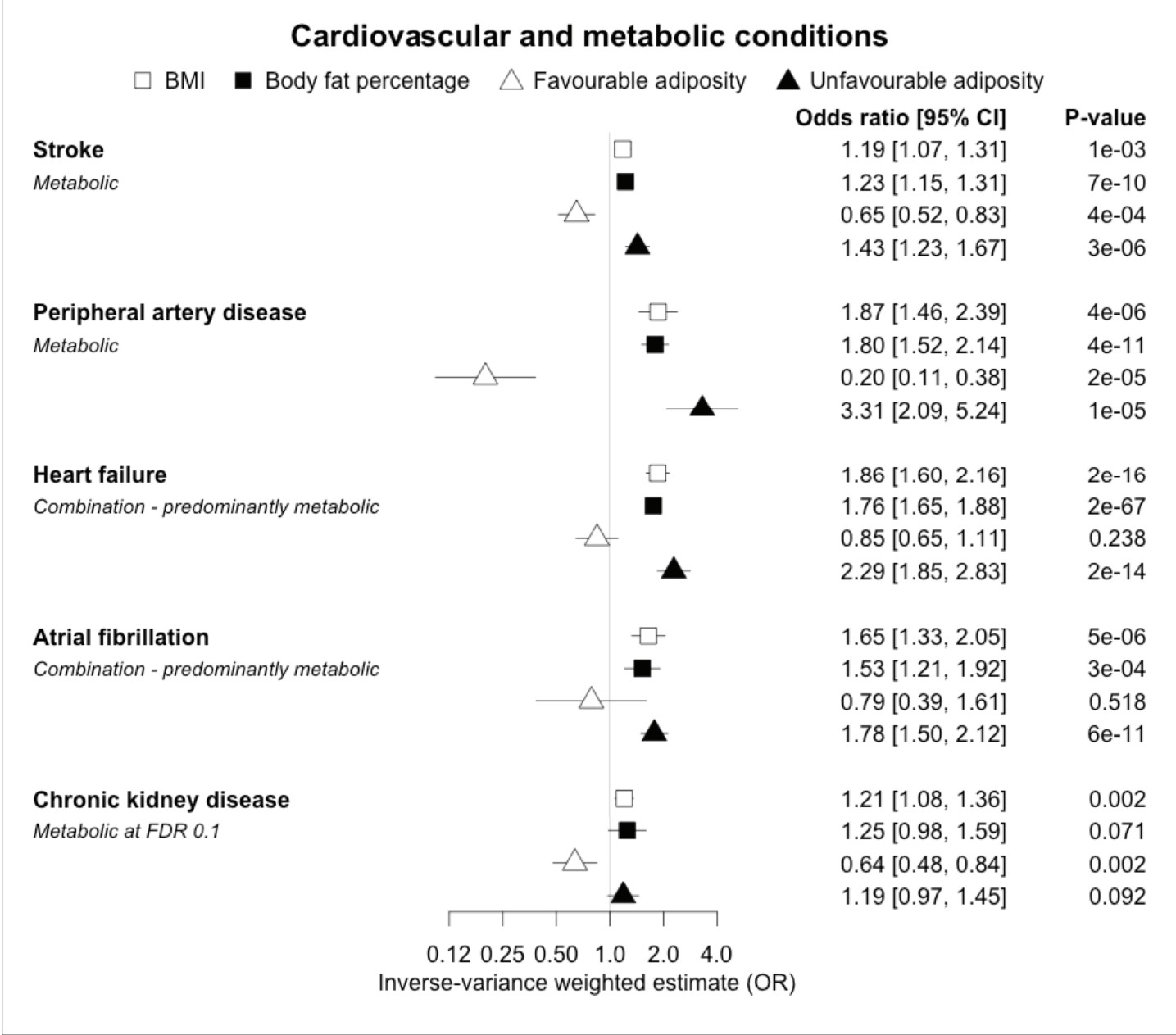

**Figure 3.** The inverse-variance weighted (IVW) two-sample MR analysis/meta-analysis of the effects of body mass index (BMI), body fat percentage (BFP), "favourable adiposity" (FA) and "unfavourable adiposity" (UFA) on stroke, peripheral artery disease, heart failure, atrial fibrillation and chronic kidney disease. The error bars represent the 95% confidence intervals of the IVW estimates in odds ratio per standard deviation change in genetically determined BMI, body fat percentage, FA and UFA. *Italics give our best interpretation of the data using the FDR 0.1 results.*

The online version of this article includes the following figure supplement(s) for figure 3:

**Figure supplement 1.** The inverse-variance weighted (IVW) two-sample MR analysis/meta-analysis of the effects of body mass index (BMI), body fat percentage (BFP), "favourable adiposity" (FA) and "unfavourable adiposity" (UFA) on sub-types of stroke.

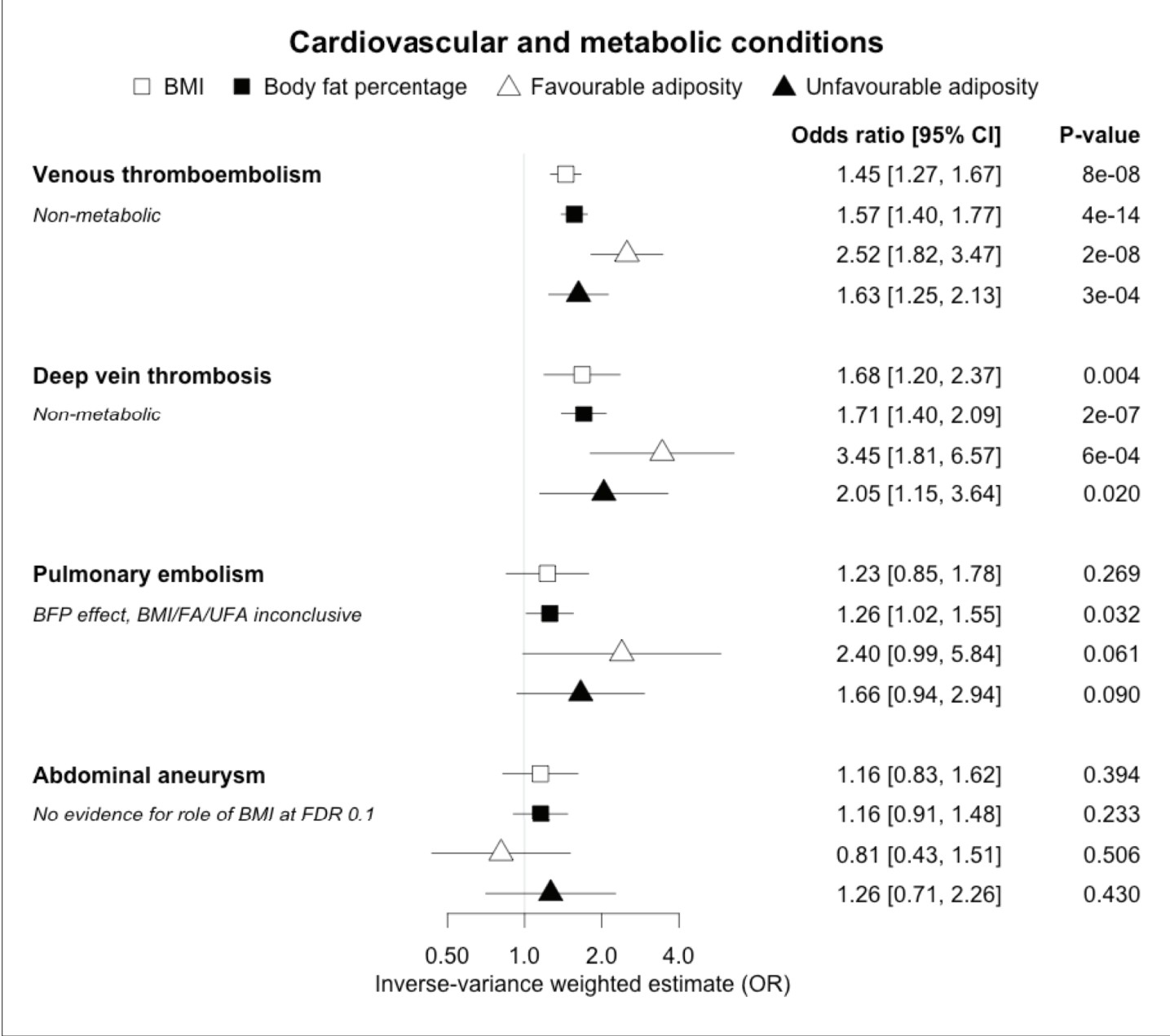

**Figure 4.** The inverse-variance weighted (IVW) two-sample MR analysis/meta-analysis of the effects of body mass index (BMI), body fat percentage (BFP), "favourable adiposity" (FA) and "unfavourable adiposity" (UFA) on venous thromboembolism, deep vein thrombosis, pulmonary embolism and abdominal aneurysm. The error bars represent the 95% confidence intervals of the IVW estimates in odds ratio per standard deviation change in genetically determined BMI, body fat percentage, FA and UFA. *Italics give our best interpretation of the data using the FDR 0.1 results.*

of breast cancer and higher risk of renal cancer, although the results from body fat percentage were less conclusive (*Figures 2–12*, *Supplementary file 1e*).

### (iv) Diseases with evidence that there is a combination of causal effects but with a predominantly non-metabolic component

When comparing the MR analyses for FA and UFA, our results suggested that some non-metabolic effect of higher adiposity is the predominant cause of the link between higher BMI and gallstones, gastro-oesophageal reflux disease, adult-onset asthma, and psoriasis (*Figures 2–12*, *Supplementary file 1e*). Our results also indicated that some non-metabolic effect of higher adiposity is causal to

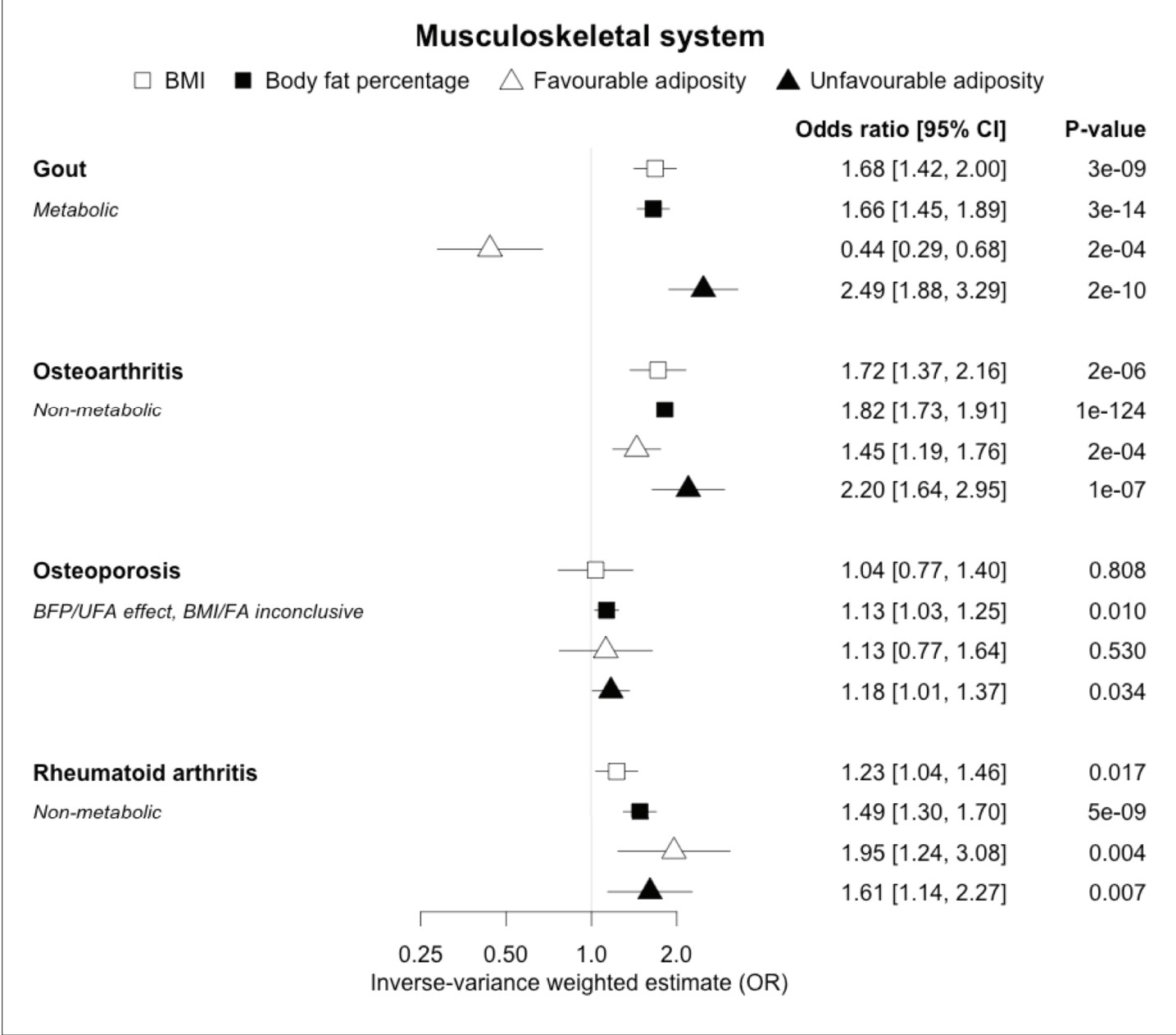

**Figure 5.** The inverse-variance weighted (IVW) two-sample MR analysis/meta-analysis of the effects of body mass index (BMI), body fat percentage (BFP), "favourable adiposity" (FA) and "unfavourable adiposity" (UFA) on gout, osteoarthritis, osteoporosis and rheumatoid arthritis. The error bars represent the 95% confidence intervals of the IVW estimates in odds ratio per standard deviation change in genetically determined BMI, body fat percentage, FA and UFA. *Italics give our best interpretation of the data using the FDR 0.1 results.*

The online version of this article includes the following figure supplement(s) for figure 5:

**Figure supplement 1.** The inverse-variance weighted (IVW) two-sample MR analysis/meta-analysis of the effects of body mass index (BMI), body fat percentage (BFP), "favourable adiposity" (FA) and "unfavourable adiposity" (UFA) on sub-types of osteoarthritis.

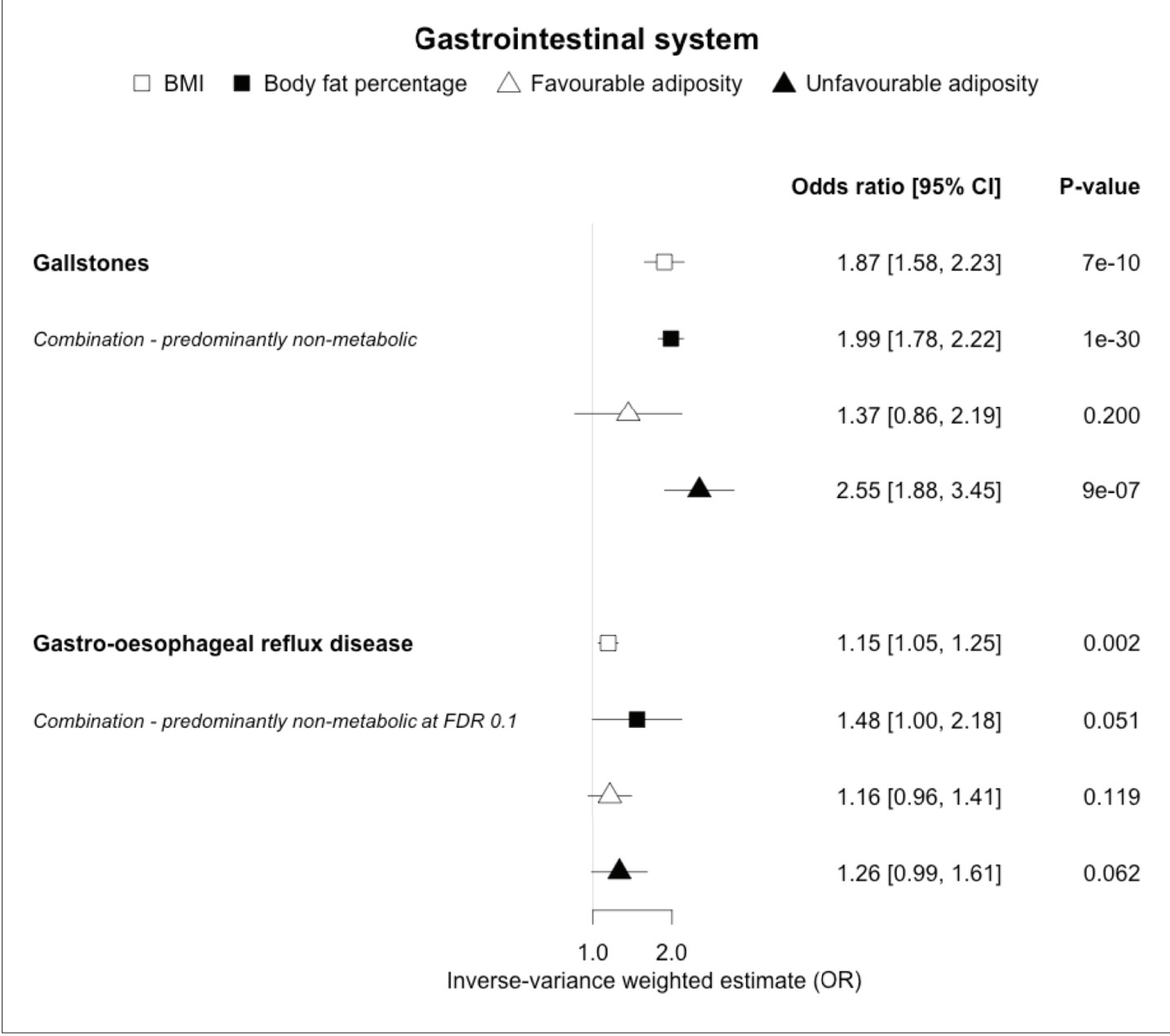

**Figure 6.** The inverse-variance weighted (IVW) two-sample MR analysis/meta-analysis of the effects of body mass index (BMI), body fat percentage (BFP), "favourable adiposity" (FA) and "unfavourable adiposity" (UFA) on gallstones and gastro-oesophageal reflux disease. The error bars represent the 95% confidence intervals of the IVW estimates in odds ratio per standard deviation change in genetically determined BMI, body fat percentage, FA and UFA. *Italics give our best interpretation of the data using the FDR 0.1 results.*

osteoporosis, although the results from BMI were less conclusive (*Figure 5*). Our results found no evidence (at p<0.05) of an effect of BMI or adiposity on child-onset asthma (*Figure 9—figure supplement 1*, *Supplementary file 1g*).

## All other disease outcomes

Fifteen disease outcomes did not fit the criteria for definitions i–iv. For five of these conditions, our MR results indicated a causal effect of higher BMI or adiposity, but results from FA and UFA were inconclusive: pulmonary embolism, depression, endometrial cancer, lung cancer, and prostate cancer (*Figures 2–12*, *Supplementary file 1e*). Additionally, we identified some evidence of a metabolic

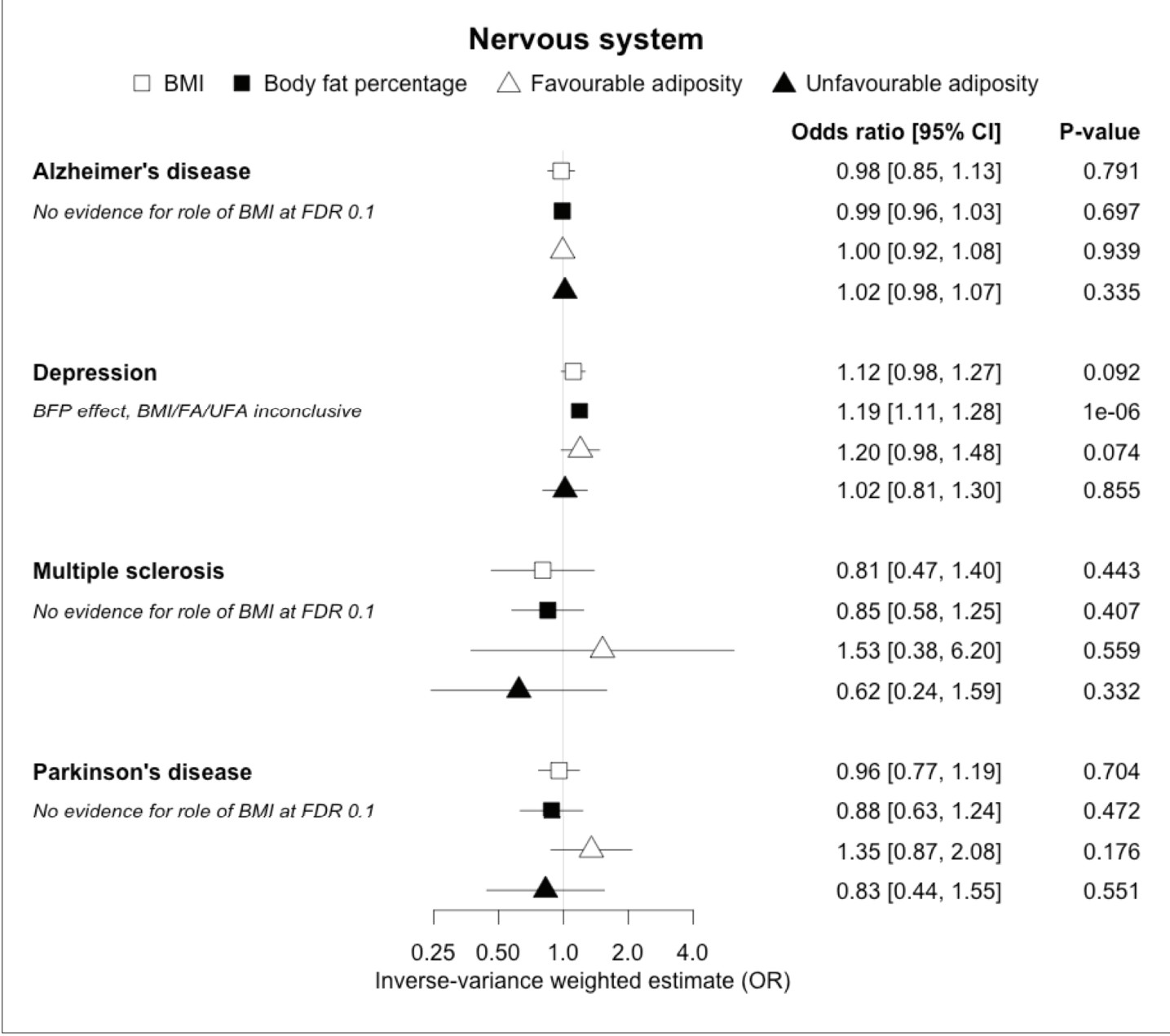

**Figure 7.** The inverse-variance weighted (IVW) two-sample MR analysis/meta-analysis of the effects of body mass index (BMI), body fat percentage (BFP), "favourable adiposity" (FA) and "unfavourable adiposity" (UFA) on Alzheimer's disease, depression, multiple sclerosis and Parkinson's disease. The error bars represent the 95% confidence intervals of the IVW estimates in odds ratio per standard deviation change in genetically determined BMI, body fat percentage, FA and UFA. *Italics give our best interpretation of the data using the FDR 0.1 results.*

effect of higher adiposity with colorectal and ovarian cancer, with the MR of FA indicating lower odds of colorectal (0.67 [0.52, 0.85]) and ovarian (0.35 [0.18, 0.70]) cancers, but MR of UFA was consistent with the null (p>0.05). For colorectal and ovarian cancer, our results were consistent when using subtypes of the conditions (*Figure 10—figure supplement 1*, *Figure 11—figure supplements 1 and 2*, *Supplementary file 1g*).

## Sensitivity analyses

Out of 82 disease outcomes (including subtypes), weighted median MR results were directionally consistent with IVW analysis for 75 diseases for BMI and 73 for body fat percentage, with 33 and 47 of

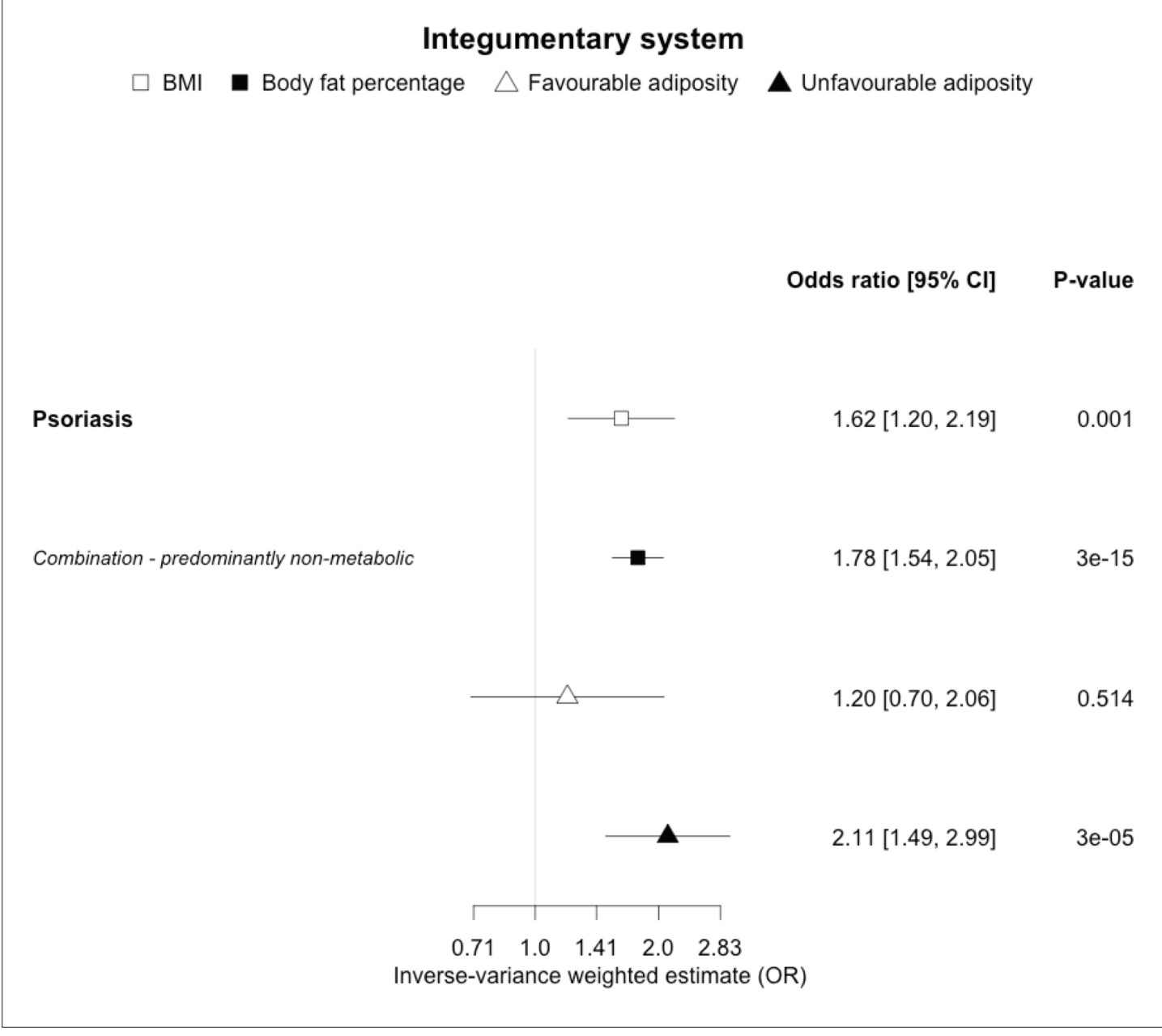

**Figure 8.** The inverse-variance weighted (IVW) two-sample MR analysis/meta-analysis of the effects of body mass index (BMI), body fat percentage (BFP), "favourable adiposity" (FA) and "unfavourable adiposity" (UFA) on psoriasis. The error bars represent the 95% confidence intervals of the IVW estimates in odds ratio per standard deviation change in genetically determined BMI, body fat percentage, FA and UFA. *Italics give our best interpretation of the data using the FDR 0.1 results.*

these having p<0.05, respectively. For FA and UFA, where sub-type colorectal cancer data was available, the total number of diseases was 87, and 76 were directionally consistent for both exposures, with 22 and 39 having p<0.05, respectively.

MR-Egger results were broadly consistent with the primary IVW MR results, indicating that pleiotropy (variants acting on the outcomes through more than one mechanism) appears to have had limited effect on our results. MR-Egger results were directionally consistent with IVW for 71 diseases for BMI and 70 for body fat percentage, with 25 and 38 of these having p<0.05, respectively. For FA and UFA, MR-Egger was directionally consistent for 60 and 67 diseases, with 6 and 15 having p<0.05, respectively (*Supplementary file 1g*). Of the 31 diseases available in the UK Biobank, the IVW analysis of these was directionally consistent with the FinnGen and/or published GWAS analysis for 28, 27, 24,

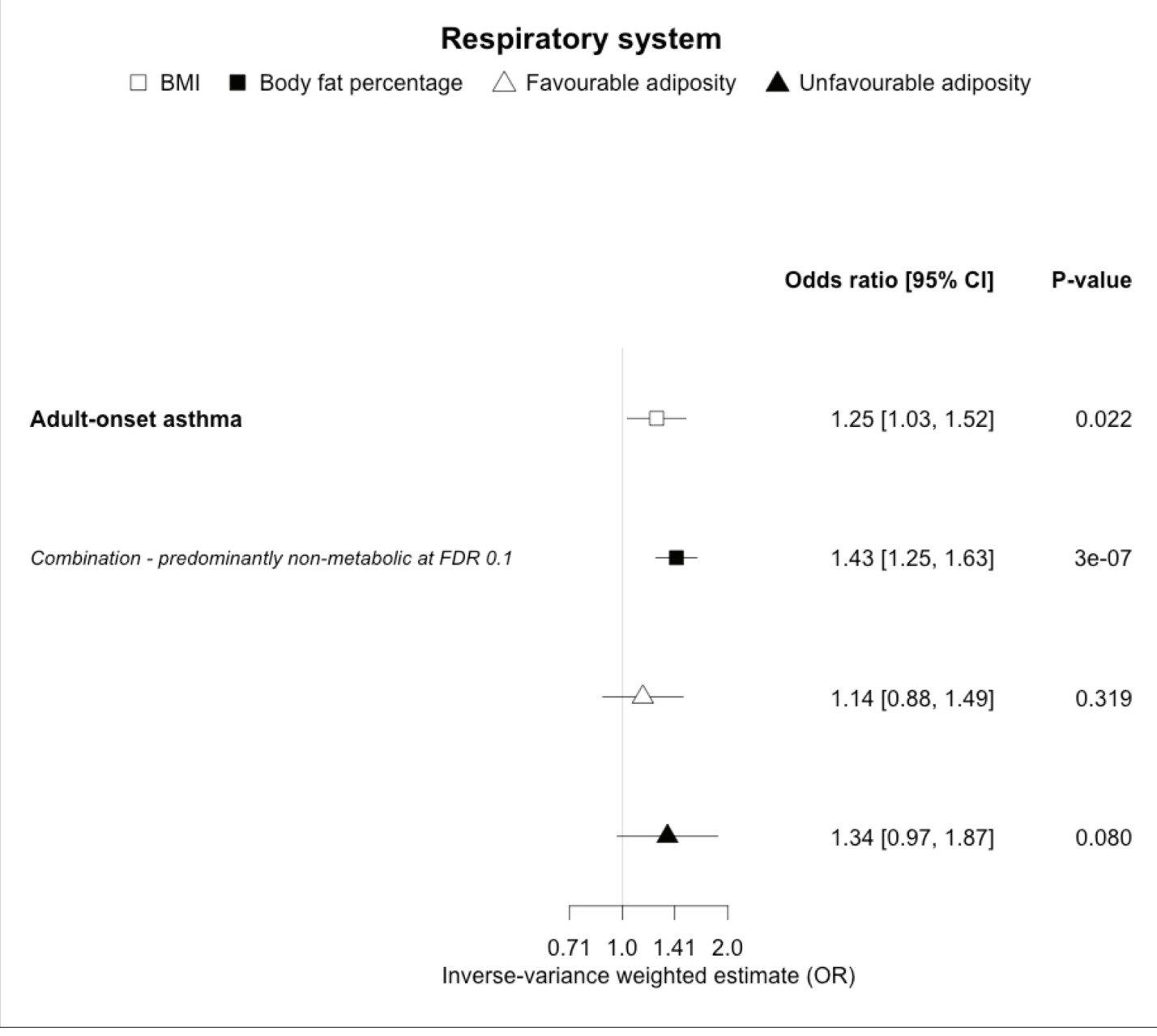

**Figure 9.** The inverse-variance weighted (IVW) two-sample MR analysis/meta-analysis of the effects of body mass index (BMI), body fat percentage (BFP), "favourable adiposity" (FA) and "unfavourable adiposity" (UFA) on adult-onset asthma. The error bars represent the 95% confidence intervals of the IVW estimates in odds ratio per standard deviation change in genetically determined BMI, body fat percentage, FA and UFA. *Italics give our best interpretation of the data using the FDR 0.1 results.*

The online version of this article includes the following figure supplement(s) for figure 9:

**Figure supplement 1.** The inverse-variance weighted (IVW) two-sample MR analysis/meta-analysis of the effects of body mass index (BMI), body fat percentage (BFP), "favourable adiposity" (FA) and "unfavourable adiposity" (UFA) on sub-types of asthma.

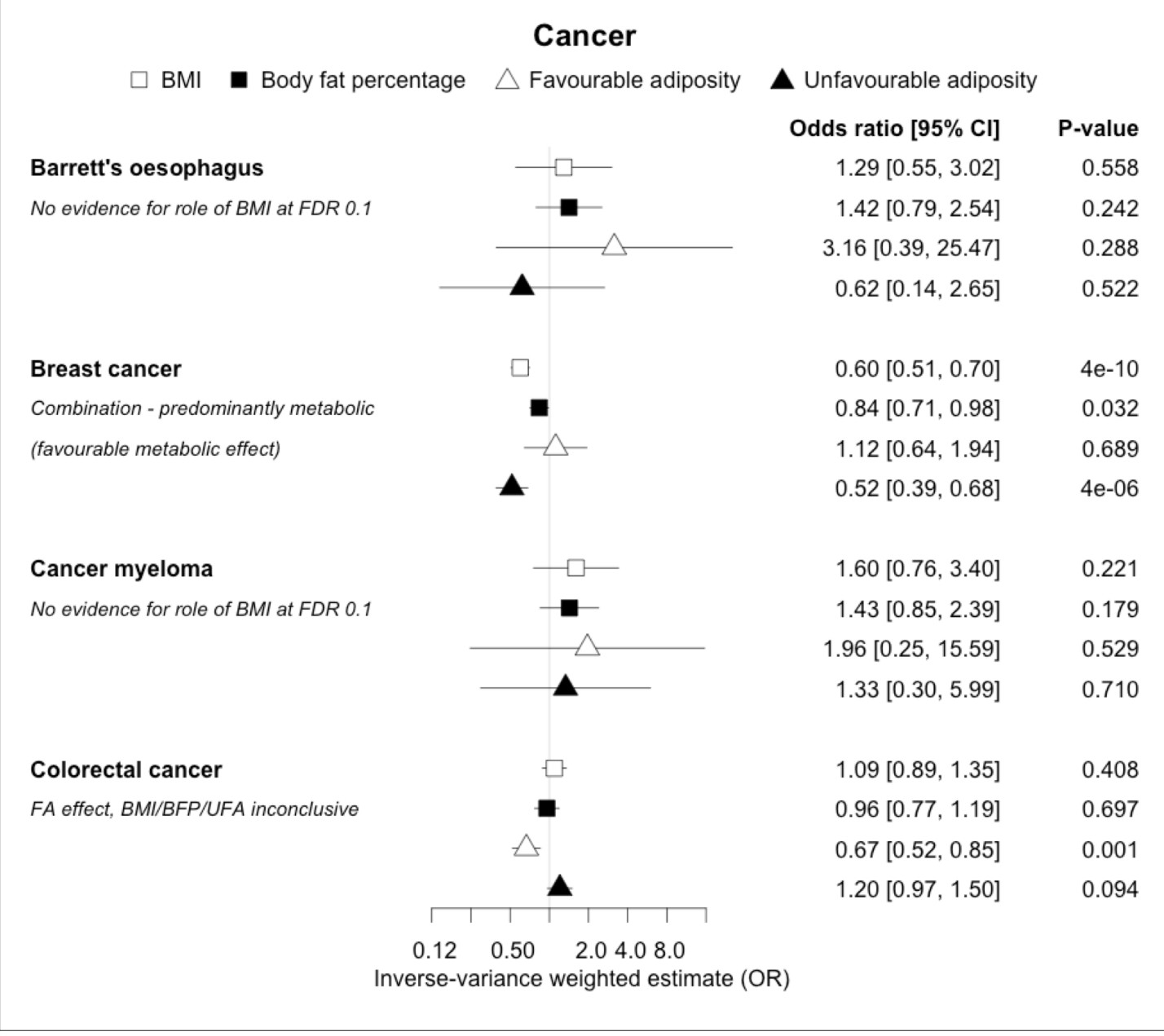

**Figure 10.** The inverse-variance weighted (IVW) two-sample MR analysis/meta-analysis of the effects of body mass index (BMI), body fat percentage (BFP), "favourable adiposity" (FA) and "unfavourable adiposity" (UFA) on Barrett's oesophagus, breast cancer, cancer myeloma and colorectal cancer. The error bars represent the 95% confidence intervals of the IVW estimates in odds ratio per standard deviation change in genetically determined BMI, body fat percentage, FA and UFA. *Italics give our best interpretation of the data using the FDR 0.1 results.*

The online version of this article includes the following figure supplement(s) for figure 10:

**Figure supplement 1.** The inverse-variance weighted (IVW) two-sample MR analysis/meta-analysis of the effects of body mass index (BMI), body fat percentage (BFP), "favourable adiposity" (FA) and "unfavourable adiposity" (UFA) on sub-types of colorectal cancer.

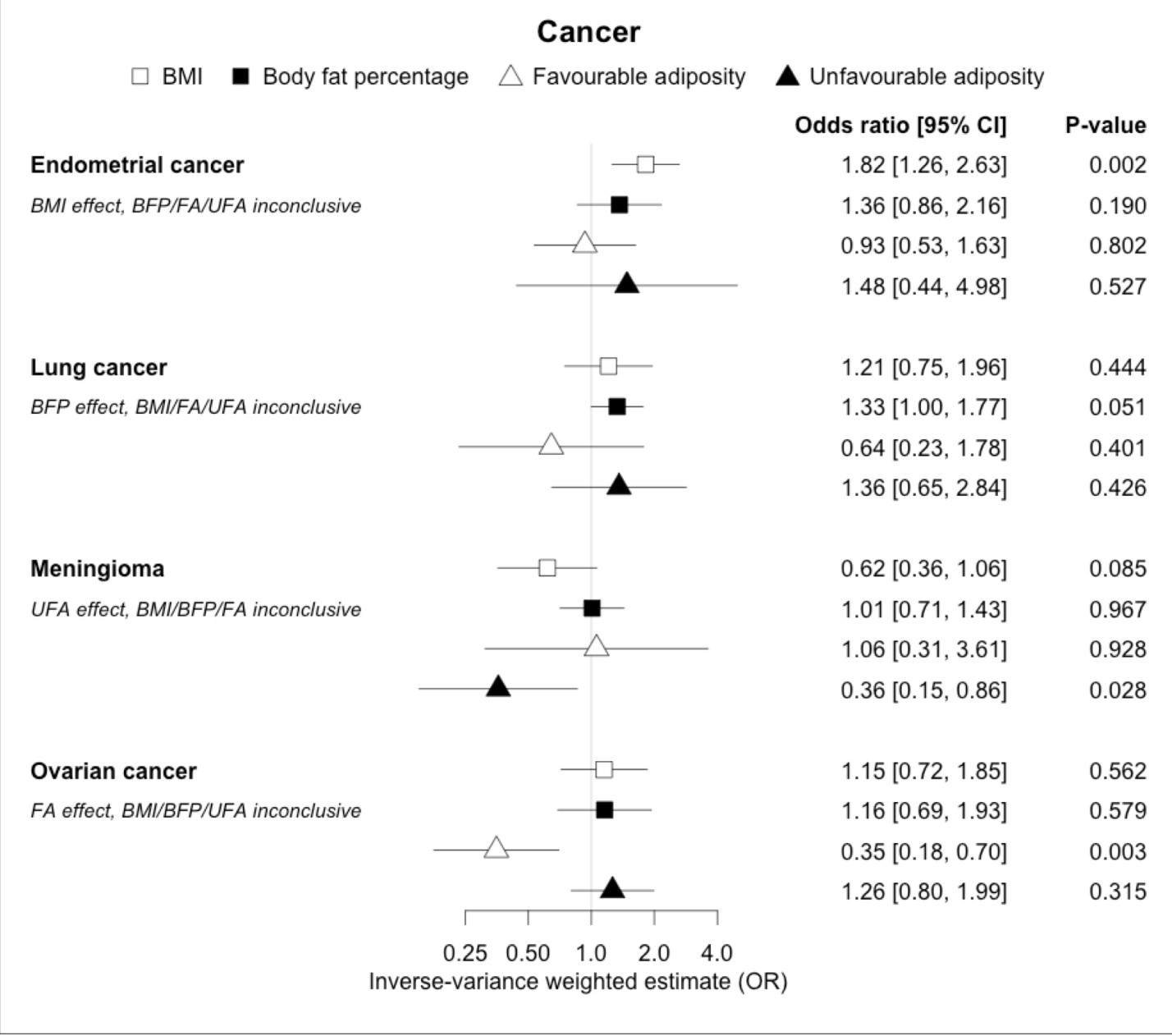

**Figure 11.** The inverse-variance weighted (IVW) two-sample MR analysis/meta-analysis of the effects of body mass index (BMI), body fat percentage (BFP), "favourable adiposity" (FA) and "unfavourable adiposity" (UFA) on endometrial and lung cancer, meningioma and ovarian cancer. The error bars represent the 95% confidence intervals of the IVW estimates in odds ratio per standard deviation change in genetically determined BMI, body fat percentage, FA and UFA. *Italics give our best interpretation of the data using the FDR 0.1 results.*

The online version of this article includes the following figure supplement(s) for figure 11:

**Figure supplement 1.** The inverse-variance weighted (IVW) two-sample MR analysis/meta-analysis of the effects of body mass index (BMI), body fat percentage (BFP), "favourable adiposity" (FA) and "unfavourable adiposity" (UFA) on 5 sub-types of ovarian cancer.

**Figure supplement 2.** The inverse-variance weighted (IVW) two-sample MR analysis/meta-analysis of the effects of body mass index (BMI), body fat percentage (BFP), "favourable adiposity" (FA) and "unfavourable adiposity" (UFA) on 4 sub-types of ovarian cancer.

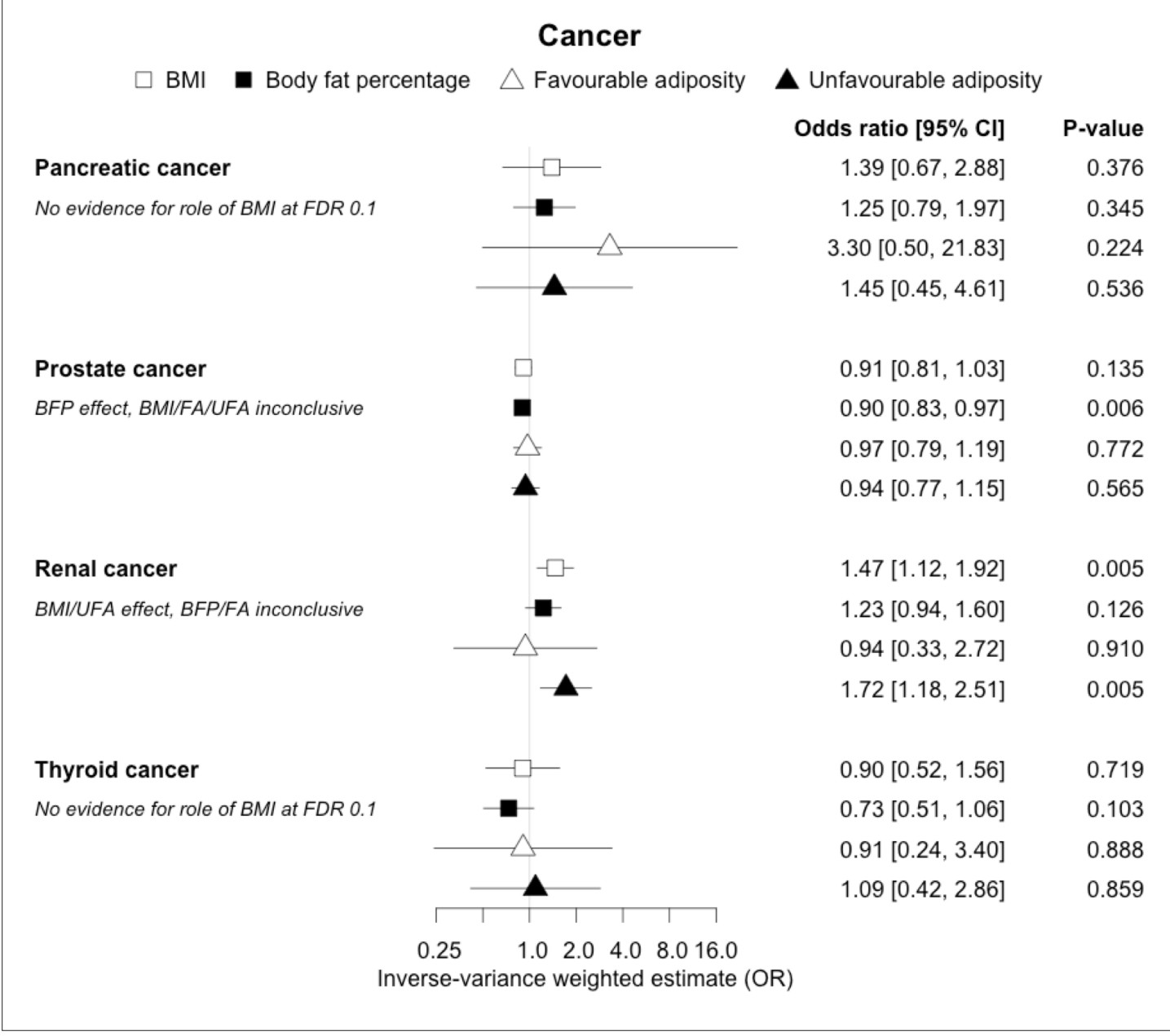

**Figure 12.** The inverse-variance weighted (IVW) two-sample MR analysis/meta-analysis of the effects of body mass index (BMI), body fat percentage (BFP), "favourable adiposity" (FA) and "unfavourable adiposity" (UFA) on pancreatic, prostate, renal and thyroid cancer. The error bars represent the 95% confidence intervals of the IVW estimates in odds ratio per standard deviation change in genetically determined BMI, body fat percentage, FA and UFA. *Italics give our best interpretation of the data using the FDR 0.1 results.*

and 27 traits for BMI, body fat percentage, FA, and UFA, respectively (**Supplementary file 1h**). Of these, 18, 21, 9, and 16 had p<0.05, respectively.

## Discussion

We used a genetic approach to understand the role of higher adiposity uncoupled from its adverse metabolic effects in mechanisms linking obesity to higher risk of disease. We first used MR to provide evidence that higher BMI was causally associated with 21 diseases, broadly consistent with those from previous studies. For the majority (17) of these diseases, our results indicated that the BMI effect was predominantly due to excess adiposity rather than a non-fat mass component to BMI. We then used

a more specific approach to test the separate roles of higher adiposity with and without its adverse metabolic effects. We provided genetic evidence that the adverse metabolic consequences of higher BMI lead to coronary artery disease, peripheral artery disease, hypertension, stroke, type 2 diabetes, polycystic ovary syndrome, heart failure, atrial fibrillation, chronic kidney disease, renal cancer, and gout, and the adverse non-metabolic consequences of higher BMI likely contribute to osteoarthritis, rheumatoid arthritis, osteoporosis, gastro-oesophageal reflux disease, gallstones, adult-onset asthma, psoriasis, deep vein thrombosis, and venous thromboembolism.

Understanding the reasons why obesity leads to disease is important in order to better advise health professionals and patients of health risks linked to obesity, whether or not they show metabolic derangements. Many previous studies have used an MR approach to support a causal role of higher BMI in disease, but here we attempted to systematically test many conditions and the role of separate components of higher BMI. We discuss some of the more notable, and potentially clinically important, results below.

## Cardiometabolic diseases

Previous studies, including those using MR, have shown that higher BMI leads to many cardiometabolic diseases (*Larsson et al., 2020*; *Riaz et al., 2018*; *Xu et al., 2020*), but our results provide additional insight into the likely mechanisms. In addition to the previously established opposing effects of metabolically FA and UFA for coronary artery disease, stroke, hypertension, and type 2 diabetes (*Martin et al., 2021*), our results confirmed similarly strong metabolic components to peripheral artery disease and chronic kidney disease. These results are consistent with the well-established adverse metabolic effects of higher BMI on these diseases (contributing to atherosclerotic effects or linked to specific haemodynamic impacts) (*Sattar and McGuire, 2018*). For two further cardiovascular conditions, heart failure and atrial fibrillation, the results were less certain. For these two conditions, the evidence of a predominantly metabolic effect of higher BMI was very clear – with the MR of UFA consistent with effects at least as strong as those for coronary artery disease. However, in contrast to the results for coronary artery disease, the MR of FA was consistent with no effect. This comparison between the effects of FA and UFA may indicate that there is a partial mechanical, or other non-metabolic component, as well as metabolic effect, perhaps mediated by excess weight of any type placing extra strain on the heart.

In contrast to the results for most of the cardiometabolic diseases, our MR analyses provided evidence for a likely non-metabolic component mediating the effect of higher BMI on venous thromboembolism and deep vein thrombosis (two closely related conditions). This finding is clinically important as it suggests that treating metabolic risk factors associated with obesity without changing weight may not reduce the risk of deep vein thrombosis in individuals with obesity. Possible mechanisms could include higher intra-abdominal pressure (due to excess fat) and slower blood circulation in the lower limbs (due to a more sedentary lifestyle secondary to obesity, or mechanical occlusion of veins) promoting clot initiation and formation (*Lorenzet et al., 2012*).

## Musculoskeletal diseases

We observed clear differences for the role of higher BMI in different musculoskeletal diseases. For gout, opposing effects of FA and UFA clearly indicated a metabolic effect. Gout is a form of inflammatory arthritis caused by the deposition of urate crystals within the joints (*Dalbeth et al., 2016*). Weight loss from bariatric surgery is associated with lower serum uric acid and lower risk of gout (*Maglio et al., 2017*). A previous MR study showed that overall obesity, but not the central location of fat, increased the risk of gout (*Larsson et al., 2018*). The protective effect of FA could be due to improved insulin sensitivity leading to less insulin-enhanced reabsorption of organic anions such as urate (*Choi et al., 2005*). In contrast to gout, our MR analysis provided evidence that a non-metabolic effect of higher adiposity is a likely cause of osteoarthritis and rheumatoid arthritis – with both FA and UFA leading to disease. For osteoarthritis, the effect of UFA was stronger than that of FA, indicating both a metabolic and non-metabolic component. This is consistent with a causal association between higher adiposity and higher risk of osteoarthritis in non-weight-bearing joints including hands (*Reyes et al., 2016*). For rheumatoid arthritis, the effects of FA and UFA were similar, suggesting the non-metabolic effect accentuating, or more readily unmasking, the autoimmune background risk, as the key BMI-related factor, although the confidence intervals were wider than those for osteoarthritis.

The UFA variants may potentially influence these conditions by load-bearing mechanisms, and tissue enrichment analysis for the FA and UFA variants previously found that FA and UFA loci are enriched for genes expressed in adipocytes and adipose tissue, and mesenchymal stem cells, respectively (*Martin et al., 2021*). For osteoporosis, we did not replicate the previous finding of a causal association between higher BMI and risk of osteoporosis (estimated by bone mineral density; *Song et al., 2020*); however, we observed a causal association between higher body fat percentage and a higher risk of osteoporosis with consistent risk increasing effects of both FA and UFA. This finding adds to the complex relationship between higher BMI and osteoporosis, where higher BMI at earlier ages may increase bone accrual, but in later years results in adverse effects.

## Gastrointestinal diseases

We observed differences in the effects of BMI when comparing the two gastrointestinal diseases, although the results are less conclusive than those for the musculoskeletal conditions. Here, our results were consistent with a predominantly non-metabolic effect contributing to the association between higher BMI and higher risk of gallstones. Higher BMI has been shown to be causally associated with higher risk of gallstones (*Yuan et al., 2021*). There are several possible mechanisms that could explain how higher BMI without its adverse metabolic effects could increase the risk of gallstones. These could include a sedentary lifestyle and gallbladder hypomotility secondary to increased abdominal fat mass (*Mathus-Vliegen et al., 2004*). Metabolic mechanisms could include hepatic de novo cholesterol synthesis (*Ståhlberg et al., 1997*; *Cruz-Monserrate et al., 2016*). For gastro-oesophageal reflux, the consistent direction and effect sizes of higher FA and UFA indicate a non-metabolic component, an effect that may be mechanical and better explained by higher central adiposity rather than overall BMI (*Green et al., 2020*).

## Other diseases

For most of the other diseases tested, it was difficult to draw firm conclusions about the role of metabolically FA and UFA. For some diseases, this was in part due to the lack of MR evidence for a role of any form of higher BMI. For example, our MR analyses provided no evidence for the role of higher BMI in the neurodegenerative diseases Alzheimer's disease, multiple sclerosis, and Parkinson's. These results are consistent with some but not all previous studies. For example, higher BMI is listed as a key risk factor for Alzheimer's disease (*Livingston et al., 2020*), although with little evidence of causality, including MR studies that failed to show an effect (*Larsson et al., 2017*; *Nordestgaard et al., 2017*). In contrast to our results, recent MR studies have indicated that higher BMI is protective of Parkinson's disease (*Noyce et al., 2017*) and causally associated with higher risk of multiple sclerosis (*Mokry et al., 2016*). For the inflammatory skin disorder psoriasis, our results indicated that both higher BMI and higher body fat percentage are causally associated with higher risk, but determining the underlying mechanism from the MR of FA and UFA was difficult. Higher BMI is a known cause of psoriasis (*Budu-Aggrey et al., 2019*; *Iskandar et al., 2015*) and weight loss is a recommended treatment (*Iskandar et al., 2015*). It is possible that both metabolic and non-metabolic pathways are driving the risk. The non-metabolic pathways could include inflammation which is one of the possible causal mechanisms (*Sbidian et al., 2017*; *Dowlatshahi et al., 2013*). Further work is required to understand if psoriasis could be effectively treated by targeting the metabolic factors alone, or whether only weight loss will benefit such patients. For cancers, our results do not provide any clear additional insight into the likely mechanisms, with potentially stronger effects for BMI and UFA compared to body fat percentage in some analyses hard to explain biologically. The reasons why higher BMI is associated with cancers is uncertain, although several MR studies indicate that the association with many is causal (*Mariosa et al., 2019*; *Vincent and Yaghootkar, 2020*), and that central adiposity may play a role (*Jarvis et al., 2016*). Exposure to higher insulin levels is a plausible mechanism, and some studies have used MR to test insulin directly (*Nead et al., 2015*; *Shu et al., 2019*; *Carreras-Torres et al., 2017b*; *Carreras-Torres et al., 2017a*; *Johansson et al., 2019*). Our MR analysis reproduced the previous finding between higher adiposity and higher risk of endometrial cancer (*Painter et al., 2016*) and renal cell carcinoma (*Johansson et al., 2019*), and lower risk of breast cancer (*Guo et al., 2016*; *Shu et al., 2019*). In contrast to previous MR studies showing a causal link between higher BMI and higher risk of prostate cancer (*Kazmi et al., 2020*; *Davies et al., 2015*), we identified a causal

association between higher body fat percentage but lower risk of prostate cancer. The relationship between higher BMI and risk of breast cancer is complicated, with MR studies indicating that higher BMI is protective of postmenopausal breast cancer (*Gao et al., 2016*). This contrasts with the epidemiological associations but could be explained by effects of childhood BMI (*Richardson et al., 2020*).

### Strengths and limitations

Our study had a number of limitations. First, we do not know all of the potential effects of the FA and UFA genetic variants on intermediary mechanisms. For example, the inflammatory profile of the FA variants needs further characterisation. However, the consistent association of the FA genetic variants with lower risk of a wide range of metabolic conditions – from type 2 diabetes where insulin resistance predominates, to stroke where atherosclerotic and blood pressure mechanisms predominate – indicates that these variants collectively represent a profile of higher adiposity and favourable metabolic factors. Second, for some diseases, we may have not had sufficient power to detect an effect of BMI or to separate the effects, and this could explain some of the null findings, especially for conditions where we might have expected an effect, such as pulmonary embolism and aortic aneurysm, but there were smaller numbers of cases available. Third, in some situations it was harder to interpret the results from the MR FA and UFA analyses, especially when one appeared to show an effect and the other did not. One possibility is that some diseases are a combination of both non-metabolic and metabolic effects. Osteoarthritis was the best example of this potential scenario because both FA and UFA increased the risk of disease, but UFA to a greater extent. However, for other diseases, it could be hard to detect a combined effect because the MR with FA could be protective (if metabolic effects predominate), increase risk (if non-metabolic effects predominate), or null (if the two have similar effects). Finally, we used an FDR of 0.1 as a guide to discussing meaningful results. We observed 21 out of the 37 outcome diseases reaching an FDR of 0.1 (based on the Benjamini–Hochberg procedure) for BMI, and 19, 11, and 20 out of the 21 diseases causally associated with BMI reaching this FDR for body fat percentage, FA, and UFA, respectively. Equivalent numbers for an FDR of 0.05 were 21, 17, 11, and 17. Excluding the five metabolic conditions used in our previous study (which were all causally associated with BMI), these results are 16, 14, 7, and 15 for an FDR of 0.1, and 16, 12, 7, and 12 for an FDR of 0.05. In addition to correcting for multiple tests, we noted that 74 of the $37 \times 4$ MR tests reached a p-value of <0.05 when we would only expect 7 by chance, suggesting many of the tests that did not reach a strict Bonferroni p<0.05 were meaningful.

In summary, we have used a genetic approach to test the separate roles of higher adiposity with and without its adverse metabolic effects. These results emphasize that many people in the community who are of higher BMI are at risk of multiple chronic conditions that can severely impair their quality of life or cause morbidity or mortality, even if their metabolic parameters appear relatively normal.

## Acknowledgements

This research has been conducted using the UK Biobank Resource under Application Number '9072' and '9055'. We acknowledge the use of the University of Exeter High-Performance Computing (HPC) facility in carrying out this work. We acknowledge use of high-performance computing funded by an MRC Clinical Research Infrastructure award (MRC Grant: MR/M008924/1).

ASTERISK: We are very grateful to Dr. Bruno Buecher without whom this project would not have existed. We also thank all those who agreed to participate in this study, including the patients and the healthy control persons, as well as all the physicians, technicians and students.

CCFR: The Colon CFR graciously thanks the generous contributions of their study participants, dedication of study staff, and the financial support from the U.S. National Cancer Institute, without which this important registry would not exist. We would like to thank the study participants and staff of the Seattle Colon Cancer Family Registry and the Hormones and Colon Cancer study (CORE Studies).

CLUE II: We thank the participants of Clue II and appreciate the continued efforts of the staff at the Johns Hopkins George W. Comstock Center for Public Health Research and Prevention in the conduct of the Clue II Cohort Study.

COLON and NQplus: We would like to thank the COLON and NQplus investigators at Wageningen University & Research and the involved clinicians in the participating hospitals.

CORSA: We kindly thank all those who contributed to the screening project Burgenland against CRC. Furthermore, we are grateful to Doris Mejri and Monika Hunjadi for laboratory assistance.

CPS-II: We thank the CPS-II participants and Study Management Group for their invaluable contributions to this research. We would also like to acknowledge the contribution to this study from central cancer registries supported through the Centers for Disease Control and Prevention National Program of Cancer Registries, and cancer registries supported by the National Cancer Institute Surveillance Epidemiology and End Results program.

Czech Republic CCS: We are thankful to all clinicians in major hospitals in the Czech Republic, without whom the study would not be practicable. We are also sincerely grateful to all patients participating in this study.

DACHS: We thank all participants and cooperating clinicians, and Ute Handte-Daub, Utz Benscheid, Muhabbet Celik, and Ursula Eilber for excellent technical assistance.

EDRN: We acknowledge all the following contributors to the development of the resource: University of Pittsburgh School of Medicine, Department of Gastroenterology, Hepatology and Nutrition: Lynda Dzubinski; University of Pittsburgh School of Medicine, Department of Pathology: Michelle Bisceglia; and University of Pittsburgh School of Medicine, Department of Biomedical Informatics.

EPIC: Where authors are identified as personnel of the International Agency for Research on Cancer/World Health Organization, the authors alone are responsible for the views expressed in this article and they do not necessarily represent the decisions, policy or views of the International Agency for Research on Cancer/World Health Organization.

The EPIC-Norfolk study: we are grateful to all the participants who have been part of the project and to the many members of the study teams at the University of Cambridge who have enabled this research.

EPICOLON: We are sincerely grateful to all patients participating in this study who were recruited as part of the EPICOLON project. We acknowledge the Spanish National DNA Bank, Biobank of Hospital Clínic–IDIBAPS and Biobanco Vasco for the availability of the samples. The work was carried out (in part) at the Esther Koplowitz Centre, Barcelona.

Harvard cohorts (HPFS, NHS, PHS): The study protocol was approved by the institutional review boards of the Brigham and Women's Hospital and Harvard T.H. Chan School of Public Health, and those of participating registries as required. We acknowledge Channing Division of Network Medicine, Department of Medicine, Brigham and Women's Hospital as home of the NHS. We would like to thank the participants and staff of the HPFS, NHS and PHS for their valuable contributions as well as the following state cancer registries for their help: AL, AZ, AR, CA, CO, CT, DE, FL, GA, ID, IL, IN, IA, KY, LA, ME, MD, MA, MI, NE, NH, NJ, NY, NC, ND, OH, OK, OR, PA, RI, SC, TN, TX, VA, WA, WY. The authors assume full responsibility for analyses and interpretation of these data.

Interval: A complete list of the investigators and contributors to the INTERVAL trial is provided in reference (32Riaz et al., 2018). The academic coordinating centre would like to thank blood donor centre staff and blood donors for participating in the INTERVAL trial.

Kentucky: We would like to acknowledge the staff at the Kentucky Cancer Registry.

LCCS: We acknowledge the contributions of Jennifer Barrett, Robin Waxman, Gillian Smith and Emma Northwood in conducting this study.

NCCCS I & II: We would like to thank the study participants, and the NC Colorectal Cancer Study staff.

NSHDS investigators thank the Biobank Research Unit at Umeå University, the Västerbotten Intervention Programme, the Northern Sweden MONICA study and Region Västerbotten for providing data and samples and acknowledge the contribution from Biobank Sweden, supported by the Swedish Research Council (VR 2017-00650).

PLCO: We thank the PLCO Cancer Screening Trial screening center investigators and the staff from Information Management Services Inc and Westat Inc. Most importantly, we thank the study participants for their contributions that made this study possible.

SEARCH: We thank the SEARCH team.

SELECT: We thank the research and clinical staff at the sites that participated on SELECT study, without whom the trial would not have been successful. We are also grateful to the 35,533 dedicated men who participated in SELECT.

UK Biobank: We would like to thank the participants and researchers UK Biobank for their participation and acquisition of data.

WHI: We thank the WHI investigators and staff for their dedication, and the study participants for making the program possible. A full listing of WHI investigators can be found at: http://www.whi.org/researchers/Documents%20%20Write%20a%20Paper/WHI%20Investigator%20Short%20List.pdf.

HY is funded by Diabetes UK RD Lawrence fellowship (grant: 17/0005594). SM and TMF are funded by the MRC (MR/T002239/1). CJB is supported by the World Cancer Research Fund (WCRF UK), as part of the World Cancer Research Fund International grant programme (IIG_2019_2009). JAB works in a unit funded by the UK MRC (MC_UU_00011/1) and the University of Bristol. EEV is supported by Diabetes UK (17/0005587) and the World Cancer Research Fund (WCRF UK), as part of the World Cancer Research Fund International grant programme (IIG_2019_2009) and works within the CRUK Integrative Cancer Epidemiology Programme (C18281/A29019).

Genetics and Epidemiology of Colorectal Cancer Consortium (GECCO): National Cancer Institute, National Institutes of Health, U.S. Department of Health and Human Services (U01 CA164930, U01 CA137088, R01 CA059045, R21 CA191312, R01201407). Genotyping/Sequencing services were provided by the Center for Inherited Disease Research (CIDR) contract number HHSN268201700006I and HHSN268201200008I. This research was funded in part through the NIH/NCI Cancer Center Support Grant P30 CA015704. Scientific Computing Infrastructure at Fred Hutch funded by ORIP grant S10OD028685.

ASTERISK: a Hospital Clinical Research Program (PHRC-BRD09/C) from the University Hospital Center of Nantes (CHU de Nantes) and supported by the Regional Council of Pays de la Loire, the Groupement des Entreprises Françaises dans la Lutte contre le Cancer (GEFLUC), the Association Anne de Bretagne Génétique and the Ligue Régionale Contre le Cancer (LRCC).

The ATBC Study is supported by the Intramural Research Program of the U.S. National Cancer Institute, National Institutes of Health.

CLUE II funding was from the National Cancer Institute (U01 CA86308, Early Detection Research Network; P30 CA006973), National Institute on Aging (U01 AG18033), and the American Institute for Cancer Research. The content of this publication does not necessarily reflect the views or policies of the Department of Health and Human Services, nor does mention of trade names, commercial products, or organizations imply endorsement by the US government.

Maryland Cancer Registry (MCR)

Cancer data was provided by the Maryland Cancer Registry, Center for Cancer Prevention and Control, Maryland Department of Health, with funding from the State of Maryland and the Maryland Cigarette Restitution Fund. The collection and availability of cancer registry data is also supported by the Cooperative Agreement NU58DP006333, funded by the Centers for Disease Control and Prevention. Its contents are solely the responsibility of the authors and do not necessarily represent the official views of the Centers for Disease Control and Prevention or the Department of Health and Human Services.

ColoCare: This work was supported by the National Institutes of Health (grant numbers R01 CA189184 [Li/Ulrich], U01 CA206110 [Ulrich/Li/Siegel/Figueireido/Colditz], 2P30CA015704-40 [Gilliland], R01 CA207371 [Ulrich/Li]), the Matthias Lackas-Foundation, the German Consortium for Translational Cancer Research, and the EU TRANSCAN initiative.

The Colon Cancer Family Registry (CCFR, https://www.coloncfr.org) is supported in part by funding from the National Cancer Institute (NCI), National Institutes of Health (NIH) (award U01 CA167551). Support for case ascertainment was provided in part from the Surveillance, Epidemiology, and End Results (SEER) Program and the following U.S. state cancer registries: AZ, CO, MN, NC, NH; and by the Victoria Cancer Registry (Australia) and Ontario Cancer Registry (Canada). The CCFR Set-1 (Illumina 1 M/1M-Duo) and Set-2 (Illumina Omni1-Quad) scans were supported by NIH awards U01 CA122839 and R01 CA143247 (to GC). The CCFR Set-3 (Affymetrix Axiom CORECT Set array) was supported by NIH award U19 CA148107 and R01 CA81488 (to SBG). The CCFR Set-4 (Illumina OncoArray

600K SNP array) was supported by NIH award U19 CA148107 (to SBG) and by the Center for Inherited Disease Research (CIDR), which is funded by the NIH to the Johns Hopkins University, contract number HHSN268201200008I. Additional funding for the OFCCR/ARCTIC was through award GL201-043 from the Ontario Research Fund (to BWZ), award 112746 from the Canadian Institutes of Health Research (to TJH), through a Cancer Risk Evaluation (CaRE) Program grant from the Canadian Cancer Society (to SG), and through generous support from the Ontario Ministry of Research and Innovation. The SFCCR Illumina HumanCytoSNP array was supported in part through NCI/NIH awards U01 CA074794 (to JDP) and /U24 CA074794 and R01 CA076366 (to PAN). The content of this manuscript does not necessarily reflect the views or policies of the NCI, NIH, or any of the collaborating centers in the Colon Cancer Family Registry (CCFR), nor does mention of trade names, commercial products, or organizations imply endorsement by the US Government, any cancer registry, or the CCFR.

COLON: The COLON study is sponsored by Wereld Kanker Onderzoek Fonds, including funds from grant 2014/1179 as part of the World Cancer Research Fund International Regular Grant Programme, by Alpe d'Huzes and the Dutch Cancer Society (UM 2012-5653, UW 2013-5927, UW2015-7946), and by TRANSCAN (JTC2012-MetaboCCC, JTC2013-FOCUS). The Nqplus study is sponsored by a ZonMW investment grant (98-10030); by PREVIEW, the project PREVention of diabetes through lifestyle intervention and population studies in Europe and around the World (PREVIEW) project which received funding from the European Union Seventh Framework Programme (FP7/2007–2013) under grant no. 312057; by funds from TI Food and Nutrition (cardiovascular health theme), a public–private partnership on precompetitive research in food and nutrition; and by FOODBALL, the Food Biomarker Alliance, a project from JPI Healthy Diet for a Healthy Life.

Colorectal Cancer Transdisciplinary (CORECT) Study: The CORECT Study was supported by the National Cancer Institute, National Institutes of Health (NCI/NIH), U.S. Department of Health and Human Services (grant numbers U19 CA148107, R01 CA81488, P30 CA014089, R01 CA197350; P01 CA196569; R01 CA201407) and National Institutes of Environmental Health Sciences, National Institutes of Health (grant number T32 ES013678).

CORSA: "Österreichische Nationalbank Jubiläumsfondsprojekt" (12511) and Austrian Research Funding Agency (FFG) grant 829675.

CPS-II: The American Cancer Society funds the creation, maintenance, and updating of the Cancer Prevention Study-II (CPS-II) cohort. This study was conducted with Institutional Review Board approval.

CRCGEN: Colorectal Cancer Genetics & Genomics, Spanish study was supported by Instituto de Salud Carlos III, co-funded by FEDER funds – a way to build Europe – (grants PI14-613 and PI09-1286), Agency for Management of University and Research Grants (AGAUR) of the Catalan Government (grant 2017SGR723), and Junta de Castilla y León (grant LE22A10-2). Sample collection of this work was supported by the Xarxa de Bancs de Tumors de Catalunya sponsored by Pla Director d'Oncología de Catalunya (XBTC), Plataforma Biobancos PT13/0010/0013 and ICOBIOBANC, sponsored by the Catalan Institute of Oncology.

Czech Republic CCS: This work was supported by the Czech Science Foundation (20-03997 S) and by the Grant Agency of the Ministry of Health of the Czech Republic (grants NV18/03/00199 and NU21-07-00247).

DACHS: This work was supported by the German Research Council (BR 1704/6-1, BR 1704/6-3, BR 1704/6-4, CH 117/1-1, HO 5117/2-1, HE 5998/2-1, KL 2354/3-1, RO 2270/8-1, and BR 1704/17-1), the Interdisciplinary Research Program of the National Center for Tumor Diseases (NCT), Germany, and the German Federal Ministry of Education and Research (01KH0404, 01ER0814, 01ER0815, 01ER1505A, and 01ER1505B).

DALS: National Institutes of Health (R01 CA48998 to M.L. Slattery).

EDRN: This work is funded and supported by the NCI, EDRN Grant (U01 CA 84968-06).

EPIC: The coordination of EPIC is financially supported by the European Commission (DGSANCO) and the International Agency for Research on Cancer. The national cohorts are supported by Danish Cancer Society (Denmark); Ligue Contre le Cancer, Institut Gustave Roussy, Mutuelle Générale de l'Education Nationale, Institut National de la Santé et de la Recherche Médicale (INSERM) (France); German Cancer Aid, German Cancer Research Center (DKFZ), Federal Ministry of Education and

Research (BMBF), Deutsche Krebshilfe, Deutsches Krebsforschungszentrum and Federal Ministry of Education and Research (Germany); the Hellenic Health Foundation (Greece); Associazione Italiana per la Ricerca sul Cancro-AIRCItaly and National Research Council (Italy); Dutch Ministry of Public Health, Welfare and Sports (VWS), Netherlands Cancer Registry (NKR), LK Research Funds, Dutch Prevention Funds, Dutch ZON (Zorg Onderzoek Nederland), World Cancer Research Fund (WCRF), Statistics Netherlands (The Netherlands); ERC-2009-AdG 232997 and Nordforsk, Nordic Centre of Excellence programme on Food, Nutrition and Health (Norway); Health Research Fund (FIS), PI13/00061 to Granada, PI13/01162 to EPIC-Murcia, Regional Governments of Andalucía, Asturias, Basque Country, Murcia and Navarra, ISCIII RETIC (RD06/0020) (Spain); Swedish Cancer Society, Swedish Research Council and County Councils of Skåne and Västerbotten (Sweden); Cancer Research UK (14136 to EPIC-Norfolk; C570/A16491 and C8221/A19170 to EPIC-Oxford), Medical Research Council (1000143 to EPIC-Norfolk, MR/M012190/1 to EPICOxford) (United Kingdom).

The EPIC-Norfolk study (https://doi.org/10.22025/2019.10.105.00004) has received funding from the Medical Research Council (MR/N003284/1 and MC-UU_12015/1) and Cancer Research UK (C864/A14136). The genetics work in the EPIC-Norfolk study was funded by the Medical Research Council (MC_PC_13048). Metabolite measurements in the EPIC-Norfolk study were supported by the MRC Cambridge Initiative in Metabolic Science (MR/L00002/1) and the Innovative Medicines Initiative Joint Undertaking under EMIF grant agreement no. 115372.

EPICOLON: This work was supported by grants from Fondo de Investigación Sanitaria/FEDER (PI08/0024, PI08/1276, PS09/02368, PI11/00219, PI11/00681, PI14/00173, PI14/00230, PI17/00509, 17/00878, PI20/00113, PI20/00226, Acción Transversal de Cáncer), Xunta de Galicia (PGIDIT07PXIB9101209PR), Ministerio de Economia y Competitividad (SAF07-64873, SAF 2010-19273, SAF2014-54453R), Fundación Científica de la Asociación Española contra el Cáncer (GCB13131592CAST), Beca Grupo de Trabajo "Oncología" AEG (Asociación Española de Gastro-enterología), Fundación Privada Olga Torres, FP7 CHIBCHA Consortium, Agència de Gestió d'Ajuts Universitaris i de Recerca (AGAUR, Generalitat de Catalunya, 2014SGR135, 2014SGR255, 2017SGR21, 2017SGR653), Catalan Tumour Bank Network (Pla Director d'Oncologia, Generalitat de Catalunya), PERIS (SLT002/16/00398, Generalitat de Catalunya), CERCA Programme (General-itat de Catalunya) and COST Actions BM1206 and CA17118. CIBERehd is funded by the Instituto de Salud Carlos III.

ESTHER/VERDI. This work was supported by grants from the Baden-Württemberg Ministry of Science, Research and Arts and the German Cancer Aid.

Harvard cohorts (HPFS, NHS, PHS): HPFS is supported by the National Institutes of Health (P01 CA055075, UM1 CA167552, U01 CA167552, R01 CA137178, R01 CA151993, R35 CA197735, K07 CA190673, and P50 CA127003), NHS by the National Institutes of Health (R01 CA137178, P01 CA087969, UM1 CA186107, R01 CA151993, R35 CA197735, K07CA190673, and P50 CA127003) and PHS by the National Institutes of Health (R01 CA042182).

Hawaii Adenoma Study: NCI grants R01 CA72520.

HCES-CRC: the Hwasun Cancer Epidemiology Study–Colon and Rectum Cancer (HCES-CRC; grants from Chonnam National University Hwasun Hospital, HCRI15011-1).

Kentucky: This work was supported by the following grant support: Clinical Investigator Award from Damon Runyon Cancer Research Foundation (CI-8); NCI R01CA136726.

LCCS: The Leeds Colorectal Cancer Study was funded by the Food Standards Agency and Cancer Research UK Programme Award (C588/A19167).

Melbourne Collaborative Cohort Study (MCCS) cohort recruitment was funded by VicHealth and Cancer Council Victoria. The MCCS was further augmented by Australian National Health and Medical Research Council grants 209057, 396414, and 1074383 and by infrastructure provided by Cancer Council Victoria. Cases and their vital status were ascertained through the Victorian Cancer Registry and the Australian Institute of Health and Welfare, including the National Death Index and the Australian Cancer Database.

Multiethnic Cohort (MEC) Study: National Institutes of Health (R37 CA54281, P01 CA033619, R01 CA063464, and U01 CA164973).

MECC: This work was supported by the National Institutes of Health, U.S. Department of Health and Human Services (R01 CA81488 to SBG and GR).

MSKCC: The work at Sloan Kettering in New York was supported by the Robert and Kate Niehaus Center for Inherited Cancer Genomics and the Romeo Milio Foundation. Moffitt: This work was supported by funding from the National Institutes of Health (grant numbers R01 CA189184, P30 CA076292), Florida Department of Health Bankhead-Coley Grant 09BN-13, and the University of South Florida Oehler Foundation. Moffitt contributions were supported in part by the Total Cancer Care Initiative, Collaborative Data Services Core, and Tissue Core at the H. Lee Moffitt Cancer Center & Research Institute, a National Cancer Institute-designated Comprehensive Cancer Center (grant number P30 CA076292).

NCCCS I & II: We acknowledge funding support for this project from the National Institutes of Health, R01 CA66635 and P30 DK034987.

NFCCR: This work was supported by an Interdisciplinary Health Research Team award from the Canadian Institutes of Health Research (CRT 43821); the National Institutes of Health, U.S. Department of Health and Human Serivces (U01 CA74783); and National Cancer Institute of Canada grants (18223 and 18226). We wish to acknowledge the contribution of Alexandre Belisle and the genotyping team of the McGill University and Génome Québec Innovation Centre, Montréal, Canada, for genotyping the Sequenom panel in the NFCCR samples. Funding was provided to Michael O. Woods by the Canadian Cancer Society Research Institute.

NSHDS: Swedish Research Council; Swedish Cancer Society; Cutting-Edge Research Grant and other grants from Region Västerbotten; Knut and Alice Wallenberg Foundation; Lion's Cancer Research Foundation at Umeå University; the Cancer Research Foundation in Northern Sweden; and the Faculty of Medicine, Umeå University, Umeå, Sweden.

OSUMC: OCCPI funding was provided by Pelotonia and HNPCC funding was provided by the NCI (CA16058 and CA67941).

PLCO: Intramural Research Program of the Division of Cancer Epidemiology and Genetics and supported by contracts from the Division of Cancer Prevention, National Cancer Institute, NIH, DHHS. Funding was provided by National Institutes of Health (NIH), Genes, Environment and Health Initiative (GEI) Z01 CP 010200, NIH U01 HG004446, and NIH GEI U01 HG 004438.

SEARCH: The University of Cambridge has received salary support in respect of PDPP from the NHS in the East of England through the Clinical Academic Reserve. Cancer Research UK (C490/A16561); the UK National Institute for Health Research Biomedical Research Centres at the University of Cambridge.

SELECT: Research reported in this publication was supported in part by the National Cancer Institute of the National Institutes of Health under Award Numbers U10 CA37429 (CD Blanke), and UM1 CA182883 (CM Tangen/IM Thompson). The content is solely the responsibility of the authors and does not necessarily represent the official views of the National Institutes of Health.

SMS and REACH: This work was supported by the National Cancer Institute grant P01 CA074184 to JDP and PAN, grants R01 CA097325, R03 CA153323, and K05 CA152715 to PAN, and the National Center for Advancing Translational Sciences at the National Institutes of Health (grant KL2 TR000421 to ANB-H).

The Swedish Low-risk Colorectal Cancer Study: The study was supported by grants from the Swedish research council; K2015-55X-22674-01-4, K2008-55X-20157-03-3, K2006-72X-20157-01-2, and the Stockholm County Council (ALF project).

Swedish Mammography Cohort and Cohort of Swedish Men: This work is supported by the Swedish Research Council /Infrastructure grant, the Swedish Cancer Foundation, and the Karolinska Institute´s Distinguished Professor Award to Alicja Wolk.

UK Biobank: This research has been conducted using the UK Biobank Resource under Application Number 8,614

VITAL: National Institutes of Health (K05 CA154337).

WHI: The WHI program is funded by the National Heart, Lung, and Blood Institute, National Institutes of Health, U.S. Department of Health and Human Services through contracts HHSN268201100046C,

HHSN268201100001C, HHSN268201100002C, HHSN268201100003C, HHSN268201100004C, and HHSN271201100004C.

## Additional information

### Competing interests
Naveed Sattar: Naveed Sattar has consulted for Afimmune, Amgen, AstraZeneca, Boehringer Ingelheim, Eli Lilly, Hanmi Pharmaceuticals, Merck Sharp & Dohme, Novartis, Novo Nordisk, Pfizer, and Sanofi; and received grant support paid to his University from AstraZeneca, Boehringer Ingelheim and Roche Diagnostics outside the submitted work. Timothy M Frayling: Tim Frayling has consulted for Boehringer Ingelheim and Sanofi and has a student supported by GSK. The other authors declare that no competing interests exist.

### Funding

| Funder | Grant reference number | Author |
| --- | --- | --- |
| Diabetes UK | 17/0005594 | Hanieh Yaghootkar |
| Medical Research Council | MR/T002239/1 | Susan Martin<br>Timothy Frayling |
| World Cancer Research Fund | IIG_2019_2009 | Caroline Bull<br>Emma E Vincent |
| Medical Research Council | MC_UU_00011/1 | Joshua A Bell |
| Diabetes UK | 17/0005587 | Emma E Vincent |
| Cancer Research UK | C18281/A29019 | Emma E Vincent |

The funders had no role in study design, data collection and interpretation, or the decision to submit the work for publication.

### Author contributions
Susan Martin, Data curation, Formal analysis, Investigation, Methodology, Software, Validation, Visualization, Writing – original draft, Writing – review and editing; Jessica Tyrrell, Conceptualization, Methodology, Project administration, Resources, Software, Supervision, Writing – review and editing; E Louise Thomas, Matthew J Bown, Lam C Tsoi, Philip E Stuart, James T Elder, Philip Law, Richard Houlston, Christopher Kabrhel, Nikos Papadimitriou, Marc J Gunter, Caroline J Bull, Joshua A Bell, Emma E Vincent, Naveed Sattar, Malcolm G Dunlop, Ian PM Tomlinson, Sara Lindström, INVENT consortium , Jimmy D Bell, Resources, Writing – review and editing; Andrew R Wood, Robin N Beaumont, Resources, Software, Writing – review and editing; Timothy M Frayling, Hanieh Yaghootkar, Conceptualization, Funding acquisition, Investigation, Methodology, Project administration, Supervision, Validation, Writing – original draft, Writing – review and editing

### Author ORCIDs
Susan Martin http://orcid.org/0000-0001-8746-0947
E Louise Thomas http://orcid.org/0000-0003-4235-4694
Naveed Sattar http://orcid.org/0000-0002-1604-2593
Jimmy D Bell http://orcid.org/0000-0003-3804-1281
Timothy M Frayling http://orcid.org/0000-0001-8362-2603
Hanieh Yaghootkar http://orcid.org/0000-0001-9672-9477

### Ethics
Human subjects: For the UK Biobank, all participants provided informed written consent and the National Research Ethics Service Committee North West-Haydock approved the study. All procedures in the UK Biobank study were conducted in accordance to the World Medical Association declaration of Helsinki ethical principles for medical research.

### Decision letter and Author response
Decision letter https://doi.org/10.7554/eLife.72452.sa1

Author response https://doi.org/10.7554/eLife.72452.sa2

## Additional files

### Supplementary files

• Supplementary file 1. Supplementary methods and Mendelian randomisation (MR) results file. (a) Characterisation of monogenic obesity, lipodystrophy, unfavourable adiposity (UFA), and favourable adiposity (FA) using body fat percentage and a selection of metabolic biomarkers. (b) Mendelian randomisation (MR) studies testing the role of obesity (usually as body mass index [BMI]) identified in literature search. (c) (i) Summary statistics of published genome-wide association studies (GWAS) used. Mean (standard deviation [SD] or range) are given for continuous study characteristics where available, mean ranges are given for meta-analyses unless otherwise specified. *Statistics represent only UK Biobank cohort of those included in meta-analysis. (c) (ii) Summary statistics of FinnGen studies used. Mean age of cases is given where available, BMI is not adjusted for, and UK Biobank is not included in these studies. ICD codes taken from hospital discharge register and/or causes of death register. (c) (iii) Summary statistics of UK Biobank studies used. Mean (SD) are given for continuous study characteristics of cases. Self-report code is from n_20002_* variable in UK Biobank. (d) The summary of 73 BMI and 696 body fat percentage genetic variants, the latter including 36 FA and 38 UFA genetic variants. Beta, SE, and p are from the GWAS of BMI and body fat percentage in UK Biobank, respectively. BMI variants were discovered using non-UK Biobank cohorts, and so some SNPs listed may have zero effect size in the UK Biobank GWAS of BMI. (e) The inverse-variance weighted two-sample MR analysis/meta-analysis of 37 identified diseases from published GWAS and/or FinnGen for BMI, body fat percentage, FA, and UFA clusters. *Italicised results are those that were interpreted – including all BMI, body fat percentage if a causal effect of BMI was indicated, and FA/UFA if a causal effect of BMI and body fat percentage was indicated.* (f) Heterogeneity statistics from random-effects meta-analysis of inverse-variance weighted MR of published GWAS and FinnGen studies. (g) (i) The inverse-variance weighted, weighted median, Egger, and penalised weighted median MR analyses for BMI using FinnGen and published GWAS. (g) (ii) The inverse-variance weighted, weighted median, Egger, and penalised weighted median MR analyses for body fat percentage using FinnGen and published GWAS. (g) (iii) The inverse-variance weighted, weighted median, Egger, and penalised weighted median MR analyses for FA using FinnGen and published GWAS. (g) (iv) The inverse-variance weighted, weighted median, Egger, and penalised weighted median MR analyses for UFA using FinnGen and published GWAS. (h) The inverse-variance weighted MR analysis of identified diseases from UK Biobank for BMI, body fat percentage, FA, and UFA clusters. PMID, PubMed ID; N, sample size; OPCS, operating procedure codes; SE, standard error; p, p-value; OR, odds ratio; 95% CI, 95% confidence interval; Q, Q-statistic; $I^2$, $I^2$-statistic; LCI, lower 95% confidence interval; UCI, upper 95% confidence interval; Intercept p, intercept p-value; $I^2$ MR-Egger, $I^2$-statistic MR-Egger.

• Transparent reporting form

### Data availability

GWAS data from the outcome diseases studied is available from links published in the original studies (Supplementary File 1ci). FinnGen data is available at: https://finngen.gitbook.io/documentation/, and the list of disease outcomes used is in Supplementary File 1cii. Individual-level UK Biobank data cannot be provided, but it is available by application to the UK Biobank: https://www.ukbiobank.ac.uk, and a list of the traits used is in Supplementary File 1ciii. Code used to conduct this analysis will be made available on GitHub after removing any sensitive information (https://github.com/susiemartin/uncoupling-bmi, copy archived at swh:1:rev:f3472762ad6cb7f313656f684e07c14b8735efe5).

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
