## [Editor Report]

The authors have conducted a robust and very comprehensive study using Mendelian randomisation to disentangle metabolic and non-metabolic effects of overweight on a long list of disease outcomes. They have tested if effects of overweight work through either or both effects for a particular condition. This is an important topic and can help us better understand how overweight influences risk of several important outcomes**.**

---

## [Decision Letter]

**Decision letter after peer review:**

Thank you for submitting your article "Disease consequences of higher adiposity uncoupled from its adverse metabolic effects using Mendelian randomisation" for consideration by *eLife*. Your article has been reviewed by 2 peer reviewers, and the evaluation has been overseen by a Reviewing Editor and a Senior Editor. The following individuals involved in review of your submission have agreed to reveal their identity: Ida Karlsson (Reviewer #1); Joris Deelen (Reviewer #2).

*Reviewer #1:*

Martin and colleagues have conducted a very comprehensive study, using Mendelian randomization to disentangle metabolic and non-metabolic effects of overweight on the risk of a long list of disease outcomes, including metabolic, gastrointestinal, neuropsychiatric, and cancer diseases. By testing causal effects of (1) BMI, (2) body fat percentage, (3) favorable adiposity, and (4) unfavorable adiposity, the authors could investigate if the effect of overweight works through metabolic or non-metabolic mechanisms, or through a combination of the two. A major strength is a robust and well structured methodology, as well as the use of large-scale data, with summary statistics published by established consortia, from FinGen, and from additional GWASs based on the UK Biobank data. The large number of outcomes tested adds substantial value to the paper, but sometimes makes the results a bit difficult to follow. Even if not all results could be interpreted in a meaningful way, the authors could indicate whether overweight influence the risk of disease through metabolic mechanisms, non-metabolic mechanisms, or both for a majority of the outcomes. This is an important topic, and can help us better understand how overweight influences risk of several important outcomes.

This is a very well conducted and well written paper, and I only suggest clarifying the text and the results, to help the reader navigate among the large number of outcomes. A smaller suggestion is to further help interpret the mechanisms of the non-metabolic effects of overweight.

1. Methods, Study design (line148-154): It is stated that MR of body fat % (step 3) was done only where a causal effect of BMI was indicated (in step 2), and MR of FA and FUA (step 4) only where a causal effect of both BMI and body fat % was indicated (in step 3). However, the Results section and the tables cover all sets of MR results for all phenotypes, regardless of findings in step 2 and 3. I don't think it's negative that all results are presented, regardless, but it was just a bit surprising to see the results, and thus confusing.

2. Methods, Disease outcomes (line 157-160): Maybe I'm missing something, but to me this does not add up: "Among the 37 identified diseases, 25 had summary GWAS data available from both a published GWAS consortium and FinnGen (23), and 12 diseases had GWAS summary data available in FinnGen only, while 31 were available in the UK Biobank". Is it because UKB is handled separately?

3. Methods: It may be helpful for the readers with a very brief introduction into FinGen. It was also not clear to me at first that you conducted additional GWASs of the disease outcomes in UKB (until I got to the MR section), that info is a bit scattered across sections. Perhaps you could add a section between study design and disease outcome, where you introduce the data sources? Including both a brief description of FinGen, UKB, and of the additional GWAS in UKB (that would also help clarify point 2).

4. Results section: I really appreciate the massive amount of work, and the large number of outcomes in this study! For obvious reasons, the results can be a bit overwhelming though, and I have the following suggestions to make it easier to follow:

a. All categories in Supplementary figure 1 (metabolic, non-metabolic, combination (predominantly metabolic) or combination (predominantly non-metabolic) causal effect) could be clarified in the beginning of the Results section. I really like Supplementary figure 1 and think the last bit of it helps when interpreting the results – perhaps it deserves being placed main paper?

b. I find Figure 1 helpful – I am not sure Table 1 is actually needed (I think all the info is also available in Figure 1?).

c. Perhaps you could also add a note about the conclusion for each outcome in Figure 1 (or as an extra column in Table 1, if you prefer to keep it)? I.e. which category of metabolic/non-metabolic/combination the results indicate.

5. Results, sensitivity analyses: Would be helpful to describe the interpretation of the sensitivity analyses, for those less familiar with MR.

6. Discussion: It would be helpful with a summary of the findings in the first section of the discussion, i.e. similar to the last paragraph. Again, I really like that you study so many different outcomes, but it is easy to forget and lose track, and a reminder here would help

7. What types of genes and pathways are the UFA SNPs involved in? Could that be noted, to help interpret those results?

*Reviewer #2 :*

The manuscript by Martin et al. reports the results from a study in which the authors tried to uncouple the metabolic and non-metabolic consequences of obesity using genetic approaches. To this end, they used Mendelian randomization and studied the effect of BMI, body fat percentage, favourable adiposity (FA) and unfavourable adiposity (UFA) on obesity-associated diseases. They managed to identify two types of obesity-associated diseases; (1) those for which metabolic effects are the likely cause of the disease and (2) those for which non-metabolic effects are the likely cause of the disease.

The strength of this study is that the authors used the results from large genetic analyses as input for their Mendelian randomization and tested their hypothesis using different approaches. However, the limitations are that described results are quite sparse (i.e. they could easily have been summarized in two short paragraphs) and that the same approach has already been used by the same authors before on a subset of the currently used phenotypes. On the other hand, the authors still managed to provide some novelty, i.e. the identification of non-metabolic obesity-associated diseases.

The described results are of interest to researchers working on obesity and could potentially impact treatment of individuals with an increased BMI. However, I am not sure how relevant the findings are for the broader community and I thus feel the manuscript is best suitable for a journal specialized in research on obesity.

I think the manuscript is well-written and the described analyses look statistically solid.

1. The Results section could be dramatically shortened. I do not see the added value of describing each set of diseases separately instead of summarizing everything in two paragraphs.

2. Moreover, it would be nice if the authors could provide some more discussion on the potential mechanisms underlying some of the observed findings (e.g. what could be the mechanisms by which the UFA genetic variants influence musculoskeletal disease (i.e. which biological pathway(s) could be responsible for this)). This could ideally be backed up by some additional data from randomized controlled trials that looked at the effect of weight reduction (i.e. perform a sub-analyses of individuals stratified by UFA), although I am not sure if such data would be publicly available. In addition, it would be good if the authors could perform an individual level allele score analysis for some of their most interesting diseases (such as osteoarthritis) within UK Biobank (as described in Davies et al. *eLife* 2019) to see if this supports their findings.

3. Last but not least, I am a bit puzzled by the number of identified variants for body fat percentage. The authors mentioned they identified 696 (independent?) genetic variants, but this seems to be a bit high (especially in comparison with the other traits). Are they sure this number is correct?

[Editors' note: further revisions were suggested prior to acceptance, as described below.]

Thank you for submitting your revised article "Disease consequences of higher adiposity uncoupled from its adverse metabolic effects using Mendelian randomisation" for consideration by *eLife*.

The reviewers have discussed their reviews with one another, and the Reviewing Editor has drafted this letter to help you prepare a revised submission.

We thank the authors who have done a great job revising the paper. We do have one final point, which we apologize for missing before:

The is some inconsistency between the results and the last sentence in the first discussion paragraph (which summarizes the results; line 532-536). The discussion sentence describes that metabolic/non-metabolic consequences of higher BMI lead to/ contribute to colorectal and ovarian cancer, cholelithiasis, and depression, when these outcomes are described as "inconclusive" with regard to FA and UFA effects in the Results section. Other outcomes, which were described as driven by metabolic/non-metabolic consequences of BMI in the results, are not included in the discussion summary sentence. The same inconsistency goes for the abstract and results. We think it would be good if the authors clarified this, but are otherwise very happy with the revised paper.

---

## [Author Response]

Reviewer #1:Martin and colleagues have conducted a very comprehensive study, using Mendelian randomization to disentangle metabolic and non-metabolic effects of overweight on the risk of a long list of disease outcomes, including metabolic, gastrointestinal, neuropsychiatric, and cancer diseases. By testing causal effects of (1) BMI, (2) body fat percentage, (3) favorable adiposity, and (4) unfavorable adiposity, the authors could investigate if the effect of overweight works through metabolic or non-metabolic mechanisms, or through a combination of the two. A major strength is a robust and well structured methodology, as well as the use of large-scale data, with summary statistics published by established consortia, from FinGen, and from additional GWASs based on the UK Biobank data. The large number of outcomes tested adds substantial value to the paper, but sometimes makes the results a bit difficult to follow. Even if not all results could be interpreted in a meaningful way, the authors could indicate whether overweight influence the risk of disease through metabolic mechanisms, non-metabolic mechanisms, or both for a majority of the outcomes. This is an important topic, and can help us better understand how overweight influences risk of several important outcomes.This is a very well conducted and well written paper, and I only suggest clarifying the text and the results, to help the reader navigate among the large number of outcomes. A smaller suggestion is to further help interpret the mechanisms of the non-metabolic effects of overweight.

We thank the reviewer for their comments, and have modified the presentation of the results to make them easier to navigate. We also added to the interpretation of the mechanisms of the non-metabolic effects of being overweight by discussing pathway and tissue enrichment in the Discussion.

1. Methods, Study design (line148-154): It is stated that MR of body fat % (step 3) was done only where a causal effect of BMI was indicated (in step 2), and MR of FA and FUA (step 4) only where a causal effect of both BMI and body fat % was indicated (in step 3). However, the Results section and the tables cover all sets of MR results for all phenotypes, regardless of findings in step 2 and 3. I don't think it's negative that all results are presented, regardless, but it was just a bit surprising to see the results, and thus confusing.

We agree and have now italicised the results in Supplementary File 1e (previously Table 1) that were / were not part of the BMI to body fat percentage to UFA and FA rationale, while still keeping all results in Supplementary File 1e for completeness.

We have now clarified in the Results section that we have presented all results in the tables for completeness:

“We focused on the MR of body fat percentage if a causal effect of BMI was indicated, and the MR of FA and UFA if a causal effect of BMI and body fat percentage was indicated, but have presented all results in Supplementary File 1e for completeness.”

2. Methods, Disease outcomes (line 157-160): Maybe I'm missing something, but to me this does not add up: "Among the 37 identified diseases, 25 had summary GWAS data available from both a published GWAS consortium and FinnGen (23), and 12 diseases had GWAS summary data available in FinnGen only, while 31 were available in the UK Biobank". Is it because UKB is handled separately?

We apologize for causing the confusion. The UK Biobank was not meant to be included here, and we have reworded this paragraph to:

"For the 37 identified diseases, 25 had summary GWAS data available from both a published GWAS consortium and FinnGen, and 12 diseases had GWAS summary data available in FinnGen only. In addition, data from 31 of the 37 diseases were available in the UK Biobank".

3. Methods: It may be helpful for the readers with a very brief introduction into FinGen. It was also not clear to me at first that you conducted additional GWASs of the disease outcomes in UKB (until I got to the MR section), that info is a bit scattered across sections. Perhaps you could add a section between study design and disease outcome, where you introduce the data sources? Including both a brief description of FinGen, UKB, and of the additional GWAS in UKB (that would also help clarify point 2).

We agree that the description of the three main sources of disease outcome data could be clearer. We have taken the reviewer’s advice and included an additional “Data sources” subsection in the Methods, including a brief introduction to FinnGen:

“We used three data sources for disease outcomes: (i) published genome-wide association studies (GWAS) and (ii) FinnGen (22) as our main results, and (iii) UK Biobank (23) as additional validation. FinnGen is a cohort of 176,899 individuals with linked medical records.”

We have moved the description of the UK Biobank that previously came under “GWAS of UK Biobank traits” further up into this new “Data sources” section.

4. Results section: I really appreciate the massive amount of work, and the large number of outcomes in this study! For obvious reasons, the results can be a bit overwhelming though, and I have the following suggestions to make it easier to follow:a. All categories in Supplementary figure 1 (metabolic, non-metabolic, combination (predominantly metabolic) or combination (predominantly non-metabolic) causal effect) could be clarified in the beginning of the Results section. I really like Supplementary figure 1 and think the last bit of it helps when interpreting the results – perhaps it deserves being placed main paper?

We have now described all four interpretations (based on Figure 1 (previously Supplementary Figure 1) – Step 5) at the beginning of the Results section by adding in combination (predominantly metabolic) and combination (predominantly non-metabolic):

“(iii) Diseases with evidence that there is a combination of causal effects but with a predominantly metabolic component. […] Here MR using the UFA genetic variants indicated that higher adiposity without its adverse metabolic consequences was likely contributing to the disease, and MR of the FA genetic variants was directionally consistent with this but FDR > 0.1.”

We agree that Supplementary Figure 1 should be a main figure, and so have set this as the new Figure 1.

b. I find Figure 1 helpful – I am not sure Table 1 is actually needed (I think all the info is also available in Figure 1?).

We agree with the reviewer and have moved Table 1 to become the new Supplementary File 1e.

c. Perhaps you could also add a note about the conclusion for each outcome in Figure 1 (or as an extra column in Table 1, if you prefer to keep it)? I.e. which category of metabolic/non-metabolic/combination the results indicate.

We thank the reviewer for their suggestion and have added into the plots in Figure 2 (previously Figure 1) and Figure 2—figure supplement 1 (previously Supplementary Figure 2) our interpretations as text with either one of the four options listed in Figure 1 (previously Supplementary Figure 1) – Step 5, “BMI/BFP/FA/UFA effect, BMI/BFP/FA/UFA inconclusive” (deleted as appropriate), or “No evidence for role of BMI at FDR 0.1”.

5. Results, sensitivity analyses: Would be helpful to describe the interpretation of the sensitivity analyses, for those less familiar with MR.

We have split the sensitivity analysis paragraph in the Methods into the UK Biobank Results section and the MR-Egger section. We have described the interpretation of the MR-Egger results by adding in the following:

“MR-Egger results were broadly consistent with the primary IVW MR results, indicating pleiotropy (variants acting on the outcomes through more than one mechanism) appears to have had limited effect on our results.”

6. Discussion: It would be helpful with a summary of the findings in the first section of the discussion, i.e. similar to the last paragraph. Again, I really like that you study so many different outcomes, but it is easy to forget and lose track, and a reminder here would help

We have moved the previous first sentence of the final paragraph in the Discussion (that lists the key findings) to become the final sentence of the first paragraph. We then replaced the first sentence of the final paragraph with:

“In summary, we have used a genetic approach to test the separate roles of higher adiposity with and without its adverse metabolic effects.”

7. What types of genes and pathways are the UFA SNPs involved in? Could that be noted, to help interpret those results?

We have now referred to the tissue/pathway-enrichment analysis of FA/UFA using DEPICT that was presented in our previous paper and have mentioned this briefly in the Discussion:

“tissue enrichment analysis for the FA and UFA variants previously found that FA and UFA loci are enriched for genes expressed in adipocytes and adipose tissue, and mesenchymal stem cells respectively (20).”

Reviewer #2 :The manuscript by Martin et al. reports the results from a study in which the authors tried to uncouple the metabolic and non-metabolic consequences of obesity using genetic approaches. To this end, they used Mendelian randomization and studied the effect of BMI, body fat percentage, favourable adiposity (FA) and unfavourable adiposity (UFA) on obesity-associated diseases. They managed to identify two types of obesity-associated diseases; (1) those for which metabolic effects are the likely cause of the disease and (2) those for which non-metabolic effects are the likely cause of the disease.The strength of this study is that the authors used the results from large genetic analyses as input for their Mendelian randomization and tested their hypothesis using different approaches. However, the limitations are that described results are quite sparse (i.e. they could easily have been summarized in two short paragraphs) and that the same approach has already been used by the same authors before on a subset of the currently used phenotypes. On the other hand, the authors still managed to provide some novelty, i.e. the identification of non-metabolic obesity-associated diseases.The described results are of interest to researchers working on obesity and could potentially impact treatment of individuals with an increased BMI. However, I am not sure how relevant the findings are for the broader community and I thus feel the manuscript is best suitable for a journal specialized in research on obesity.

We thank the public reviewers for their comments, and have worked to reduce the size of the Results section to make it easier to navigate for the reader.

I think the manuscript is well-written and the described analyses look statistically solid.1. The Results section could be dramatically shortened. I do not see the added value of describing each set of diseases separately instead of summarizing everything in two paragraphs.

We have revised the Results section to be grouped by interpretation (based on the 4 result outcomes in Step 5 of Figure 1 (previously Supplementary Figure 1)) rather than disease type, as well as a fifth subsection for all other disease outcomes that do not meet the criteria for the 4 main definitions. This included deleting the “Diseases in this category included…” and “For example… [results with odds ratios, etc.]” sentences previously included. This shortens the Results section considerably.

2. Moreover, it would be nice if the authors could provide some more discussion on the potential mechanisms underlying some of the observed findings (e.g. what could be the mechanisms by which the UFA genetic variants influence musculoskeletal disease (i.e. which biological pathway(s) could be responsible for this)). This could ideally be backed up by some additional data from randomized controlled trials that looked at the effect of weight reduction (i.e. perform a sub-analyses of individuals stratified by UFA), although I am not sure if such data would be publicly available. In addition, it would be good if the authors could perform an individual level allele score analysis for some of their most interesting diseases (such as osteoarthritis) within UK Biobank (as described in Davies et al. eLife 2019) to see if this supports their findings.

We have added a clearer sentence in the Discussion about UFA potentially being load-bearing as well as discussing tissue enrichment analysis results presented in our previous paper:

“The UFA variants may potentially influence these conditions by load-bearing mechanisms, and tissue enrichment analysis for the FA and UFA variants previously found that FA and UFA loci are enriched for genes expressed in adipocytes and adipose tissue, and mesenchymal stem cells respectively (20).”

We do not have data from randomised controlled trials available to us to conduct the additional analysis described – but agree with the reviewer that it would be a valuable addition if possible.

We do not feel that an individual-level allele score analysis will add anything further to our study – other studies show that two-sample MR has more power, and we would have to run unweighted individual-level allele score analysis as both the genetic instruments and individual-level data would come from the UK Biobank (causing sample overlap). Most importantly, the UK Biobank has far fewer diseases cases than the published GWAS where only summary statistics are available. Finally, Davies et al. *eLife* 2019 do not find anything novel from running this as a sensitivity analysis – their results were consistent with their primary two-sample MR analysis.

3. Last but not least, I am a bit puzzled by the number of identified variants for body fat percentage. The authors mentioned they identified 696 (independent?) genetic variants, but this seems to be a bit high (especially in comparison with the other traits). Are they sure this number is correct?

This number is correct – it is larger because these independent SNPs were identified using UK Biobank, whereas the BMI variants were discovered using non-UK Biobank cohorts. We have made this clearer in the legend of Supplementary File 1d (previously Supplementary Table 4):

“BMI variants were discovered using non-UK Biobank cohorts”.

[Editors' note: further revisions were suggested prior to acceptance, as described below.]

We thank the authors who have done a great job revising the paper. We do have one final point, which we apologize for missing before:The is some inconsistency between the results and the last sentence in the first discussion paragraph (which summarizes the results; line 532-536). The discussion sentence describes that metabolic/non-metabolic consequences of higher BMI lead to/ contribute to colorectal and ovarian cancer, cholelithiasis, and depression, when these outcomes are described as "inconclusive" with regard to FA and UFA effects in the Results section. Other outcomes, which were described as driven by metabolic/non-metabolic consequences of BMI in the results, are not included in the discussion summary sentence. The same inconsistency goes for the abstract and results. We think it would be good if the authors clarified this, but are otherwise very happy with the revised paper.

We thank the reviewers for their comments and for spotting this inconsistency in the description of the results within the manuscript. We have corrected for this by removing colorectal and ovarian cancer and depression from the first summary paragraph in the Discussion, and instead listing all those conditions with evidence of a metabolic/non-metabolic causal effect or a combination of causal effects with a predominantly metabolic/non-metabolic component. We have also updated the results in the Abstract to match.